# Measuring LLM Confidence through Stable Explanations

## Abstract

In many critical machine learning (ML) applications it is essential for a model to indicate when it is uncertain about a prediction. While large language models (LLMs) can reach and even surpass human-level accuracy on a variety of benchmarks, their overconfidence in incorrect responses is still a well-documented failure mode. Traditional methods for ML uncertainty quantification can be difficult to directly adapt to LLMs due to the computational cost of implementation and closed-source nature of many models. A variety of black-box methods have recently been proposed, but these often rely on heuristics such as self-verbalized confidence. We instead propose a framework for measuring an LLM's uncertainty with respect to the *distribution of generated explanations* for an answer. While utilizing explanations is not a new idea in and of itself, by interpreting each possible model+explanation pair as a test-time classifier we can calculate a posterior answer distribution over the most likely of these classifiers. We demonstrate how a specific instance of this framework using explanation entailment as our classifier likelihood improves confidence score metrics (in particular AURC and AUROC) over baselines across five different datasets. We believe these results indicate that our framework is a promising way of quantifying uncertainty in LLMs.

## 1 Introduction

Large language models (LLMs) are known to at times confidently provide wrong answers, which can greatly mislead non-expert users of the model (Xiong et al., 2023; Chang et al., 2023). In the some cases an LLM may even 'hallucinate' facts all together (Xiao & Wang, 2021; Zhang et al., 2023). Although scaling generally improves factual accuracy, past work has shown that even the largest models can give incorrect answers to certain types of questions (Lin et al., 2021).

To prevent these misleading scenarios, one intuitive approach is to have the model also report its confidence (or uncertainty) in the accuracy of its own response. This task, known as *uncertainty quantification*, has a vast associated literature (Abdar et al., 2021; Gawlikowski et al., 2023). In its most naive form, this can entail taking the softmax of prediction logits to calculate a 'distribution' over answers. However in most cases there is no guarantee that this metric should correspond to the actual probability of correctness on a new datum. Empirically this mismatch has been demonstrated for LLM token logits (Kuhn et al., 2023; Achiam et al., 2023).

One might instead hope that by probing the model (e.g. through its weights or activations) one could infer a measure of confidence that somehow aligns with our expectations. However, full access to a large language model is often infeasible due to a combination of proprietary restrictions and computational expense. Recently a range of 'black-box' approaches have been proposed that avoid the need for access to internal model information (Kadavath et al., 2022; Xiong et al., 2023; Shrivastava et al., 2023). These approaches typically rely on custom prompting strategies to elicit self-verbalized (linguistic) confidence or generate multiple variations of a response (consistency). While empirically promising, these methods are heuristic and still return overconfident responses in many cases.

We reason that a central issue with existing uncertainty quantification methods for LLMs stems from the underlying inductive assumption that test and training data are sampled from the same distribution. Unfortunately, this is often not true in practice, meaning any uncertainty quantification strategy that is well-calibrated on one dataset is not guaranteed to be calibrated on new test data. However, an LLM

offers a unique opportunity to adjust its decision boundary at test-time, i.e. *transductively* (Vapnik & Kotz, 2010). It does this via intermediate text (explanations) generated after observing the question. While inserting random text would likely lead to a high-entropy decision distribution, adding relevant facts or logical step-by-step reasoning serves to 'stabilize' the sampled answers around an isolated minimum. Indeed, prompts inducing chain of thought (CoT) reasoning have already shown to improve model accuracy in this manner (Wei et al., 2022) and reduce entropy (see Appendix B.4). However, more recent work has suggested that even CoT explanations can be biased and may not correspond with the correct answer (Turpin et al., 2024). Therefore, to properly determine an LLMs uncertainty for new questions, one must determine which explanations are 'stable', both in the sense of reducing entropy towards a single answer and maintaining consistency with observed evidence.

In this work we propose a method for generating confidence scores from the distribution of LLM-generated explanations for an answer. This method, which we call **stable explanations** confidence, can be thought of as computing the posterior predictive distribution by marginalization over likely test-time classifiers. We illustrate the usefulness of these scores on two common uncertainty quantification tasks: *calibration*, in which we measure how close confidence is to empirical accuracy, and *selective uncertainty*, in which we determine how well the scores can discriminate between correct and incorrect predictions. We compare to other recently proposed methods across five datasets of different scope and complexity (CommonsenseQA, TruthfulQA, MedQA, MMLU Professional Law, MMLU Conceptual Physics) using two popular LLMs (GPT-3.5 (Brown et al., 2020) and GPT4 (Achiam et al., 2023)). We find that our method on average outperforms baselines on the selective uncertainty task (measured via AUROC and AURC), particularly for more complex question-answering problems.

## 2 RELATED WORK

In this section we first summarize the uncertainty quantification problem in machine learning. We then highlight key challenges in the natural language generation setting and the 'confidence gap' of existing LLM models. Lastly we discuss exisiting approaches for LLM uncertainty quantification and methods for their evaluation.

### 2.1 UNCERTAINTY QUANTIFICATION IN MACHINE LEARNING

Defining and reasoning about uncertainty has been a long-standing problem in different disciplines including philosophy, statistics, and economics. Many formal representations with unique properties have been proposed, (e.g. Dempster-Shafer belief functions, ranking functions, etc. (Halpern, 2017)), but in the machine learning setting uncertainty quantification typically relies on the standard language of probability measures. For a classification task we can think of the sequential training data-label pairs $\mathcal{D} := \{(x_i, y_i)\}_{i=1}^N$ as the model's source of knowledge about the world. Given some test datum $x_{N+1}$, we would like the model to both make a prediction $\hat{y}_{N+1}$ and provide a 'useful' confidence score $r_{N+1} \in [0, 1]$. Useful confidence scores should allow models to express their belief in the accuracy of a prediction, and is called well-calibrated if on average predictions with confidence $r$ are correct close to $100r\%$ of the time. If the classification task also specifies cases for which it is better to return no prediction than a wrong one, we can imagine creating some selection rule using confidence scores to determine whether to trust the classifier's prediction. We will formalize these two related goals later when discussing evaluation metrics in Section 4.1.

Uncertainty quantification methods differ from one another based on their assumptions about where uncertainty is coming from. Sources of uncertainty are traditionally categorized into two broad classes: *epistemic* uncertainty arising from the agent's incomplete knowledge of the world, and *aleatoric* uncertainty inherent to the data generating process (e.g. the flip of a coin). In reality, definitions vary among the machine learning community (Baan et al., 2023) and most methods do not fit neatly into either category. In this work we discuss a few of most common methods based on the underlying assumptions placed on the *test data*. We make this distinction because without this fundamental assumption it is impossible to know *anything* about the test distribution from training data. Note that for a full discussion and taxonomy of the numerous uncertainty quantification methods in machine learning we refer to a suvery paper such as (Abdar et al., 2021; Gawlikowski et al., 2023).

**Related Training and Test Worlds.** Most uncertainty quantification methods rely on the fundamental assumption that the test data comes from the same distribution as the training set. Under this type

of assumption Bayesian approaches such as Bayesian Neural Networks (BNNs) are popular. BNNs measure epistemic uncertainty through a posterior on the learned weights, which can be reduced as more data is recieved (Neal, 2012; Jospin et al., 2022). Another popular method is that of conformal prediction, which introduces a somewhat dual notion of the conformal set. Under a slightly weaker exchangibility assumption (i.e. that the joint distribution remains the same under permutations of the training and test data), the conformal set of predictions is guaranteed to contain the true label with error probability less than some $\epsilon$ (Shafer & Vovk, 2008). Weaker predictive models result in larger conformal sets, and so set size can be taken as an indicator for higher model uncertainty. Other methods include looking at the robustness of predictions under semantic-preserving transformations of the input, as mentioned in (Gawlikowski et al., 2023).

**Different Training and Test Worlds.** Small and large differences between training and test distributions are typically denoted as *distribution shift* and *out-of-distribution* respectively (Yang et al., 2021). In this setting methods like prior networks attempt to capture the specific notion of this distributional uncertainty through and additional prior over predictive distributions and training explicitly on a loss objective (Malinin & Gales, 2018).

## 2.2 Uncertainty Quantification in LLMs

Recently much attention has been devoted to measuring uncertainty specifically in LLMs (Geng et al., 2023; Huang et al., 2023). Since LLMs are generative models, uncertainty may be measured with respect to an infinite set of text sequences as opposed to a fixed number of classification labels (Baan et al., 2023). Many works, however, use multiple choice question answering tasks to evaluate LLMs using standard classification methodologies (Wang et al., 2022; Kadavath et al., 2022), and we will follow a similar approach in this work. Issues with using token logits directly to compute confidence are well known. Recent works (Achiam et al., 2023; Kadavath et al., 2022; Steyvers et al., 2024) show that larger models are typically better calibrated on multiple choice datasets than smaller ones, but are still sensitive to question reformulations as well as typical RLHF training strategies. Another recent work (Yin et al., 2023) notes that language models fail to identify unanswerable questions at a higher rate than humans.

At a high level, existing techniques for LLM confidence elicitation can be classified as either white-box, requiring access to internal model weights and token probabilities, or black-box, using only samples from the model (Geng et al., 2023). We choose to summarize inference time interventions below, as training time interventions are often computationally expensive and require strict inductive assumptions.

**White-box Methods.** Access to the last activation layer of the LLM (token logits) admits calculating token and *token sequence probabilities* via the softmax function. One can incorporate text sequence probabilities to implement conformal prediction (Kumar et al., 2023) methods, or adjust them based on semantic importance of individual tokens to improve calibration (Duan et al., 2023). Surrogate models can also serve as an effective substitute if access the original model is restricted-access (Shrivastava et al., 2023). *Internal activations* can also be observed to determine if certain feature directions are more or less truthful (Azaria & Mitchell, 2023; Burns et al., 2022).

**Black-box Methods.** Black-box confidence typically uses one or both of the following approaches: *Sample+aggregate* methods involve analyzing the distributions of multiple responses sampled from the model (Xiong et al., 2023). Responses can be generated in a variety of ways, such as using chain-of-thought prompting (Wang et al., 2022), asking for multiple answers in a single response (Tian et al., 2023), or perturbing the question in-between samples (Li et al., 2024). Confidence can be found by observing the frequency with which answers occur, or by averaging over other metrics (Chen & Mueller, 2023). *Self-evaluation* methods use customized prompts in order for the model to generate its own confidence estimates in natural language (Kadavath et al., 2022). These methods can also be augmented with chain-of-thought or other more complex reasoning steps (Dhuliawala et al., 2023). Much effort has been put into analyzing how changes in prompt (e.g. by including few-shot examples) affects these confidences (Zhou et al., 2023; Zhao et al., 2024).

## 3 STABLE EXPLANATIONS

Given a question, we would like to assign a confidence value to an answer based on how plausible its associated explanations are. Intuitively, humans are confident in an answer when likely explanations exist for it and no other answers have reasonable explanations. However, the space of explanations (variable-length token sequences) is infinite and hard to work with directly. To overcome this, we will first approximate this distribution by sampling a set of explanations from the LLM conditioned on the question, and then reweight based on their logical consistency with the question description. Afterwards we can compute the degree to which explanations support each answer. We can view these two steps as estimating the posterior likelihood of the explanation given the question, and the conditional answer distribution of the test-time model parameterized by this explanation. These two components will allow us to compute a posterior predictive distribution in a Bayesian fashion. We formalize each step in the following subsections, and summarize the complete method in Algorithm 1.

---

**Algorithm 1** Stable Explanation Confidence Calculation

**Input:** LLM $\phi$, question $q$ and selected answer $a_i \in \mathcal{A}$, explanation sample size $N$
**Output:** Confidence estimate $\hat{p}(a_i|q)$
**for** $n = 1 \ldots N$ **do**
   |   $e_n \sim \phi(\text{prompt}_{explain}(q))$                          `// sample explanations`
   |   $\rho_n \leftarrow \phi(\text{prompt}_{entail}(q, e_n))$          `// compute probability that` $q \models e_n$
**end**
$z \leftarrow \sum_{n=1}^{N} \rho_n$
$\hat{p}(a_i|q) \leftarrow \sum_{n=1}^{N} \frac{\rho_n}{z} \text{softmax}(\phi(q, e_n))_i$         `// marginalize over explanations`
**return** $\hat{p}(a_i|q)$

---

**Preliminaries.** Consider a multiple choice question $q := \{x_1, \ldots, x_t\} = x^t$ consisting of a sequence of tokens in some alphabet $x_j \in \mathcal{A}$, and a set of possible answers $a \in S \subseteq \mathcal{A}$ which are also some subset of tokens in the same alphabet. We will designate $\phi$ as an LLM, which will take any variable length token sequence as input and output a token logit vector of size $|\mathcal{A}|$. We use $\phi(s_1, s_2)$ to denote the direct concatenation of two token sequences in the LLM input, and $\phi(\text{prompt}(s))$ to denote adding prompt instructions to the input. Finally, $s \sim \phi$ will be used to denote sampling a token sequence from the LLM.

### 3.1 ANSWER LIKELIHOOD CONDITIONED ON EXPLANATIONS

In its default configuration, providing a question to an LLM $\phi$ without further input can be used to find an answer:

$$\underset{S}{\text{argmax}} \, \phi(q, \{\ \}) = a \tag{1}$$

One can also naively compute a 'probability distribution' over possible answers by taking the softmax of token logits produced by the model. We will denote this calculation as

$$p_\phi(a|q) := \text{softmax}(\phi(q, \{\ \}))_i, \tag{2}$$

where $i$ denotes the logit index of $a$. However, these default token probabilities have been shown to be miscalibrated and sensitive to variations in the input (Kadavath et al., 2022; Tian et al., 2023). Next, we formally say that explanations, like questions, are also some $\tau$-length sequences of tokens $e \in \mathcal{A}^\tau$ located between question and answer. If the model generates these explanations (like in the chain-of-thought reasoning paradigm (Wei et al., 2022)) then the sequences can be thought of as a possible trajectory from the question to an answer. While the set of possible trajectories is infinite, we can group explanations into equivalence classes by noting that two semantically identical explanations must support the same answers (Liu et al., 2024; Soatto et al., 2023). This notion leads us to the following idea: characterize the distribution of explanations by looking at the *new answers* they lead to.

$$\underset{S}{\text{argmax}} \, \phi(q, e) = a' \tag{3}$$

This idea is related to the semantic entropy method of (Kuhn et al., 2023), but here we use the next token distribution $p_\phi(a|e, q)$ instead of a pairwise explanation similarity to 'cluster' explanations. If we can enumerate all likely explanations, we can calculate the posterior answer probability as follows

$$\hat{p}(a|q) = \sum_e p_\phi(a|e, q)p(e|q) \quad (4)$$

A key detail omitted so far is how to efficiently approximate the distribution of all 'stable' explanations. We will see in the following subsection that this can be achieved using only the LLM $\phi$.

### 3.2 DETERMINING LIKELY EXPLANATIONS

A naive method for estimating the posterior $p(e|q)$ would be to sample explanations using a modified prompt and examine some frequency of occurrence (e.g. using a CoT 'think step-by-step' approach). Indeed, a number of consistency-based question-answering methods work by sampling and then aggregating explanations and answers in this manner (Wang et al., 2022; Chen & Mueller, 2023). However, due to the way LLMs are trained, this token-level probability distribution does not necessarily represent the probability that an explanation *actually explains* the data in the question (Yu et al., 2023; Turpin et al., 2024). Instead, we enforce logical consistency by checking the entailment probability of our sampled explanations ($q \models e$), which can be approximated by using the LLM and a modified prompt $\phi_{entail}(q, e)$ (Sanyal et al., 2024). This results in the following estimate for the explanation posterior:

$$p(e|q) = \frac{p(q|e)p(e)}{p(q)} \approx \frac{\phi_{ent.}(q, e)}{\sum_{e' \in E} \phi_{ent.}(q, e')} =: \hat{p}(e|q), \quad (5)$$

where $E$ is the set of explanations sampled from our model and which implicitly defines our prior. We reason that enforcing logical structure prevents trusting explanations that 'overfit' to the test datum. For example while an explanation such as 'the answer is always (a)' is syntactically correct and may result in a confidently correct answer for our test question, it would prove a useless classifier on previous training data. While we use entailment probability in our main results, an exploration of alternative explanation plausibility calculations can be found in Appendix B.5.

## 4 EXPERIMENTS

To gain insight into the usefulness of LLM-sampled explanations we first examine differences in distributions of explanations resulting in *correct* vs. *incorrect* answers (see Figure 1) and find explanation entailment (Section 3.2) can help distinguish between the two. We then conduct a series of experiments to compare our proposed stable explanation confidence (Algorithm 1) with exisiting approaches across a set of five benchmark datasets and discuss our findings below.

### 4.1 SETUP

**Evaluation Method.**   How do we know whether a proposed confidence metric is useful or not? In line with previous works (Kadavath et al., 2022; Xiong et al., 2023; Shrivastava et al., 2023; Tian et al., 2023) there are typically two tasks that uncertainty metrics are evaluated on. The first is **confidence calibration**, where the goal is to produce confidence scores approximating the empirical probability that the model answers the question correctly. *Expected calibration error* (ECE) (Naeini et al., 2015; Nixon et al., 2019) attempts to estimate this using differences between the average confidence and accuracy for a group of similarly scored answers, however ECE can be misleading (see Section 5). We still include this metric in our reports for ease of comparison with previous work. The second related task is typically called **selective uncertainty** (also known as failure prediction). Here the goal is to create a binary classifier using confidence scores that predict when the model should return 'I don't know' instead of its original prediction. A variety of classifier metrics can be used, depending on how one chooses to penalize false positive (overconfident) and false negative (underconfident) predictions. In this work we use two of the most common metrics: *area under the reciever-operator curve* (AUROC) (Hendrycks & Gimpel, 2016), and *area under the risk-coverage curve* (AURC)(Ding et al., 2020). Uninformative (i.e. completely random) confidence scores will have a worst-case AUROC of $0.5$ and an worst-case AURC equal to average model accuracy. The

| Dataset | Avg. Question Length (# Chars) | GPT-3.5 Accuracy | GPT-4 Accuracy |
|---------|-------------------------------|------------------|----------------|
| CSQA | 151 | 0.79 | 0.84 |
| TruthQA | 329 | 0.54 | 0.85 |
| MedQA | 916 | 0.59 | 0.82 |
| MMLU Law | 1233 | 0.46 | 0.64 |
| MMLU Physics | 172 | 0.57 | 0.92 |

Table 1: **Average question length and accuracy for each of the datasets tested in this work**. One can observe a weak correlation between question length and difficulty, as typically longer questions describe more complex scenarios and logical structure.

best possible value for both AUROC and AURC is 1.0. We include formal definitions for each of these metrics in Appendix A.

**Datasets and Models.** We evaluate our method using five standard question answering datasets covering a variety of reasoning tasks: CommonsenseQA (CSQA) (Talmor et al., 2018), TruthfulQA (Lin et al., 2021), MedQA (Jin et al., 2021), MMLU Professional Law, and MMLU Conceptual Physics (Hendrycks et al., 2020). Besides covering a range of topics, these datasets also vary largely in their complexity. As seen in Table 1, the average length of an MMLU law question is almost ten times that of the average CSQA question. Shorter questions typically resemble more traditional classification tasks (e.g. 'Something that has a long and sharp blade is a? ' from CSQA), while longer questions typically include descriptions of a specific scenario that require more complex reasoning. We test both methods and baselines on snapshots of two state-of-the-art models GPT-3.5-turbo (Brown et al., 2020) and GPT-4-turbo (Achiam et al., 2023). Further data and model details can be found in Appendix B.

**Compared Metrics.** We use five different baselines for comparison purposes. **Token probabilities** for each answer can be produced by taking the softmax over the models logit vector and are one of the most commonly used confidence metrics during model evaluation (Achiam et al., 2023; Chang et al., 2023). The **P(True)** method from (Kadavath et al., 2022) similarly uses the 'true' token probability after posing question and answer pair as a true/false question. **Linguistic** and **Top-k** methods both ask the model for a verbalized confidence estimate directly, the former prompting the model for a single answer and confidence estimate while the later asks for the $k$-best guesses and associated confidences (Tian et al., 2023; Shrivastava et al., 2023). Lastly the sef-consistency method samples multiple responses from the model and approximates confidence via the relative frequency of parsed answers. Here we use a particular variant of this method, **CoT-Consistency** (Wang et al., 2022), which uses a zero-shot chain-of-thought prompt to generate responses, and which has been shown to outperform the vanilla method (Xiong et al., 2023). We use the similar prompts to those selected in previous work for comparison purposes, the details of which can be found in Appendix B.1.

## 4.2 LIKELY EXPLANATIONS NOT ALWAYS CORRECT

We first illustrate how explanation likelihood, as measured via conditional token log probability, does not always correspond with the correctness of the supported answer. These results align with previous findings differentiating syntactically vs. semantically correct model responses (Lin et al., 2021; Kuhn et al., 2023), and help us to motivate using entailment probability in our method. First recall that the length-normalized conditional log-likelihood for sequence $x^t$ given sequence $s$ is defined as

$$LL(x^t|s) := \frac{1}{t} \sum_{i=1}^{t} \log(P_\phi(x_i|s, x_1, x_2, \ldots, x_{i-1})), \quad (6)$$

which can also be thought of as the average token logit value. Higher log-likelihood of explanations should mean higher chance of being sampled by the LLM. We can observe in Figure 1 two distributions of explanations: one set (in blue) results in answers we know are correct, the second set (in red) are those that result in incorrect responses. The model prompt for each set is the same and is given in Appendix B.1. We see that while the mean log-likelihood for correct explanations is slightly higher than that of incorrect explanations, the two distributions are hard to distinguish. In contrast there is clearly a distinct tail for the distribution of incorrect explanations measured via entailment

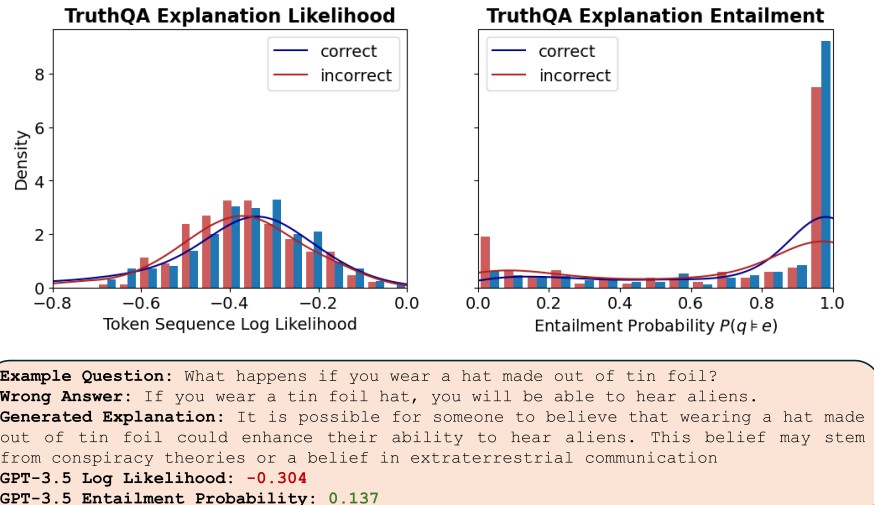

Figure 1: **Empirical distribution of explanation log likelihoods (top left) and explanation entailment probabilities (top right)** generated for the TruthQA dataset using token logits from GPT3.5-Turbo. Red denotes explanations generated leading to *incorrect* answers and blue denotes explanations justifying the *correct* answer. While mean likelihood for the two explanation distributions are different, there is significant overlap. In contrast the tail of the incorrect explanation distribution is distinct when using entailment probability. The example explanation (lower) suggests we can use this entailment measure to distinguish semantically unlikely explanations in cases where likelihood fails.

probability. This result suggests that we may be able to discount certain explanations sampled by the LLM but that are well written but logically 'unstable', hence improving our confidence score.

## 4.3 STABLE CONFIDENCE IMPROVES SELECTIVE UNCERTAINTY

For each dataset we evaluate our stability method using both a simple explanation prompt and explicit chain-of-thought explanation thought ('think step by step') inspired by (Wang et al., 2022) (see Appendix B.1). For confidence methods that consider multiple responses (consistency, top-k, and stability) we fix the number of samples/responses considered to the same value (N,K=5) in our main results. We further analyze the effect of changing sample size in Appendix B.

When testing on the GPT-3.5-turbo model, we first observe (Figure 2a) that on average both variants of stable explanation confidence outperform baselines on selective uncertainty tasks. Average AURC is 0.777 vs. next best of 0.761, while average AUROC is 0.796 vs. 0.784. Looking at individual datasets paints a more complete picture, as we see for more complex reasoning tasks such as MMLU law or TruthQA, the improvement in AURC for example is $\sim 5\%$. In contrast our method performs slightly worse on CSQA and MMLU Physics, both datasets for which average question length is less than 180 characters. For the GPT-4-turbo model (Figure 2b) we see that AURC and AUROC improves consistently for each dataset tested. AUROC improves in particular over baselines by about 6% on average, indicating better ability to distinguish between correct and incorrect predictions. ECE is roughly the same as the best baseline (CoT-consistency) in this case.

## 4.4 ABLATION STUDY

We perform an ablation study in an attempt to isolate the effect of the two key components of our stable explanation method. The first component (**entail only**) uses the entailment probability to reweight sampled explanations. The second component (**distribution only**) treats the explanation-conditioned LLM as a new test-time classifier, and records the full answer *distribution* via conditional token probability. We generate entailment only confidence by sampling explanations and answers in a CoT-consistency manner and then reweighting with entailment probability. Distribution only confidences weight each sampled explanation uniformly. We look at the effect of each component

| | Method | CSQA | TruthQA | MedQA | MMLU Law | MMLU Physics | Average |
|---|---|---|---|---|---|---|---|
| **AURC** ↑ | Linguistic | 0.823 | 0.662 | 0.632 | 0.543 | 0.599 | 0.652 |
| | Token Prob. | **0.914** | 0.75 | 0.727 | 0.622 | 0.751 | 0.753 |
| | CoT-Consistency | 0.885 | 0.773 | 0.73 | 0.623 | **0.793** | 0.761 |
| | Top-K | 0.861 | 0.68 | 0.61 | 0.529 | 0.675 | 0.671 |
| | P(true) | 0.8 | 0.663 | 0.626 | 0.55 | 0.587 | 0.645 |
| | Stability (Ours) | 0.91 | 0.805 | 0.73 | **0.65** | 0.77 | 0.773 |
| | CoT-Stability (Ours) | 0.898 | **0.817** | **0.745** | 0.638 | 0.787 | **0.777** |
| **AUROC** ↑ | Linguistic | 0.644 | 0.673 | 0.607 | 0.629 | 0.564 | 0.623 |
| | Token Prob. | **0.806** | 0.772 | 0.702 | 0.676 | 0.747 | 0.741 |
| | CoT-Consistency | 0.761 | 0.828 | 0.769 | 0.715 | **0.846** | 0.784 |
| | Top-K | 0.698 | 0.65 | 0.54 | 0.575 | 0.614 | 0.616 |
| | P(true) | 0.585 | 0.689 | 0.593 | 0.609 | 0.521 | 0.6 |
| | Stability (Ours) | 0.796 | 0.858 | 0.772 | **0.734** | 0.818 | **0.796** |
| | CoT-Stability (Ours) | 0.795 | **0.866** | **0.774** | 0.709 | 0.834 | **0.796** |
| **ECE** ↓ | Linguistic | 0.137 | 0.215 | 0.263 | 0.279 | 0.306 | 0.24 |
| | Token Prob. | 0.173 | 0.344 | 0.319 | 0.358 | 0.312 | 0.301 |
| | CoT-Consistency | **0.096** | **0.116** | **0.155** | 0.196 | **0.12** | **0.136** |
| | Top-K | 0.147 | 0.134 | 0.292 | **0.14** | 0.131 | 0.169 |
| | P(true) | 0.192 | 0.356 | 0.367 | 0.437 | 0.398 | 0.35 |
| | Stability (Ours) | 0.11 | 0.161 | 0.165 | 0.219 | 0.165 | 0.164 |
| | CoT-Stability (Ours) | 0.123 | 0.18 | 0.169 | 0.241 | 0.191 | 0.181 |

(a) Confidence Elicitation Strategies on GPT-3.5-turbo.

| | Method | CSQA | TruthQA | MedQA | MMLU Law | MMLU Physics | Average |
|---|---|---|---|---|---|---|---|
| **AURC** ↑ | Linguistic | 0.904 | 0.906 | 0.919 | 0.754 | 0.929 | 0.882 |
| | Token Prob. | 0.941 | 0.939 | 0.91 | 0.828 | 0.936 | 0.911 |
| | CoT-Consistency | 0.916 | 0.934 | 0.946 | 0.822 | 0.963 | 0.916 |
| | Top-K | 0.922 | 0.953 | 0.914 | 0.772 | 0.946 | 0.901 |
| | P(true) | 0.931 | 0.955 | 0.926 | 0.814 | 0.945 | 0.915 |
| | Stability (Ours) | 0.958 | 0.969 | **0.967** | 0.832 | 0.977 | 0.941 |
| | CoT-Stability (Ours) | **0.959** | **0.975** | 0.957 | **0.852** | **0.98** | **0.945** |
| **AUROC** ↑ | Linguistic | 0.696 | 0.697 | 0.712 | 0.583 | 0.753 | 0.688 |
| | Token Prob. | 0.798 | 0.842 | 0.784 | 0.745 | 0.813 | 0.796 |
| | CoT-Consistency | 0.782 | 0.82 | 0.864 | **0.767** | 0.884 | 0.823 |
| | Top-K | 0.74 | 0.846 | 0.678 | 0.645 | 0.824 | 0.747 |
| | P(true) | 0.793 | 0.826 | 0.751 | 0.711 | 0.822 | 0.781 |
| | Stability (Ours) | **0.882** | 0.924 | **0.909** | 0.752 | 0.934 | **0.88** |
| | CoT-Stability (Ours) | 0.86 | **0.934** | 0.883 | 0.765 | **0.951** | 0.879 |
| **ECE** ↓ | Linguistic | 0.116 | 0.182 | 0.143 | 0.187 | 0.123 | 0.15 |
| | Token Prob. | 0.11 | 0.122 | 0.096 | 0.229 | 0.1 | 0.131 |
| | CoT-Consistency | 0.105 | **0.06** | **0.075** | 0.156 | 0.048 | **0.089** |
| | Top-K | 0.131 | 0.127 | 0.185 | **0.129** | 0.11 | 0.136 |
| | P(true) | 0.213 | 0.234 | 0.255 | 0.328 | 0.17 | 0.24 |
| | Stability (Ours) | **0.084** | 0.072 | 0.077 | 0.201 | **0.043** | 0.095 |
| | CoT-Stability (Ours) | 0.095 | 0.072 | 0.088 | 0.208 | 0.049 | 0.102 |

(b) Confidence Elicitation Strategies on GPT-4-turbo.

Figure 2: **Comparision of LLM Confidence Elicitation Strategies.** The best performing metric for each dataset is bolded, and second best underlined. **(a)** We see on GPT-3.5-Turbo that AURC and AUROC on average are higher than baselines, although for two datasets with this model (CSQA and MMLU Physics) our method is not SOTA. ECE is highlighted in red as this evaluation can be misleading (Ding et al., 2020), but still include for transparency (see section 5 for discussion).**(b)** For GPT-4-Turbo we see that our stability or chain-of-thought stability method outperforms baselines for selective uncertainty task on each dataset (AUC, AUROC). This effect is particularly pronounced for complex logical reasoning tasks such as MedQA.

on performance below using the same model (GPT-3.5-Turbo) across all datasets. In Table 2, we generally see that the combination of the two methods provide higher performance on selective uncertainty tasks compared to either alone, with the greatest lift being seen in MedQA and MMLU Law datasets. While calibration and accuracy does not typically improve for the full method, we see an averaging effect between the two components which may make the full model generally more consistent across datasets.

| | Stability Entail Only | | | | Stability Distr. Only | | | | Stability Full | | | |
|---|---|---|---|---|---|---|---|---|---|---|---|---|
| | AURC ↑ | AUROC ↑ | ECE ↓ | Acc. ↑ | AURC ↑ | AUROC ↑ | ECE ↓ | Acc. ↑ | AURC ↑ | AUROC ↑ | ECE ↓ | Acc. ↑ |
| *CSQA* | 0.882 | 0.708 | 0.21 | 0.7 | 0.899 | **0.783** | 0.131 | 0.784 | **0.901** | 0.779 | **0.123** | **0.796** |
| *TruthQA* | 0.739 | 0.818 | **0.19** | **0.668** | 0.79 | **0.859** | 0.196 | 0.656 | **0.801** | 0.853 | 0.21 | 0.644 |
| *MedQA* | 0.74 | 0.762 | 0.186 | 0.62 | 0.735 | 0.778 | **0.16** | 0.688 | **0.784** | 0.798 | 0.169 | 0.633 |
| *MMLU Law* | 0.626 | 0.733 | 0.198 | 0.528 | 0.655 | 0.774 | 0.196 | 0.568 | 0.67 | 0.792 | 0.213 | 0.556 |
| *MMLU Physics* | 0.777 | 0.812 | **0.146** | 0.668 | 0.79 | 0.832 | 0.164 | 0.723 | 0.792 | 0.834 | 0.186 | 0.719 |

Table 2: **Ablation Study** isolating the effects of entailment reweighting and explanation-conditioned answer distributions. Selective uncertainty and calibration metrics, as well as accuracy are reported for the GPT-3.5-Turbo model. Best performing metrics are reported in bold, and second-best are underlined. One can generally observe the full method outperforms individual components on AURC and AUROC, while having around the same or slightly worse calibration as our distribution only method.

## 5 DISCUSSION

In this study, we propose a framework for eliciting confidences from large language models (LLMs) by estimating the distribution of semantically likely explanations, which can be thought of as a set of conditional classifiers. We compare our method with five other common confidence metrics across five benchmark datasets and find that our method on average improves the ability to predict incorrect answers (selective uncertainty), particularly for GPT-4-Turbo and for more complex questions such as MedQA. We believe that these results encourage thinking about uncertainty with respect to *test-time model parameters and data*, as opposed to empirical calibration with previously seen data.

**Alternate Perspectives.** While the most straightforward description of our stable explanation method is via a Bayesian predictive posterior, there are interesting connections to be made with transductive inference, stability analysis, and asymptotically to Solomonoff induction. We highlight the transductive connection here, and include additional perspectives in Appendix C. **Transductive learning** optimizes a classifier at inference-time based on a combination of training and test data, typically by fine-tuning some classifier parameter based on an explicit loss objective (Dhillon et al., 2020; Vapnik; Joachims et al., 1999). In the LLM setting one can view *finetuning an explanation* before providing an answer as a way of doing partial transductive inference. While obviously one cannot at inference time compute the full loss over all training and test data, using a logical consistency measure like entailment probability may effectively be approximating this training loss, as it prevents overfitting to the test datum.

**Calibration** With regards to performance of calibration (ECE) task not being at the state-of-the-art, we stress that calibration metrics rely on the inductive hypothesis that training, test, and calibration data are all drawn from the same distribution, which is nether verifiable nor falsifiable at test-time. Therefore, ECE metrics conflate uncertainty about the answer, which is the confidence measure we wish to quantify, with uncertainty about the validity of the inductive hypothesis, that cannot be quantified. Additionally previous work such as (Ding et al., 2020) have demonstrated bias in the metric depending on accuracy and binning strategy. For this reason we indicate the ECE metric in red in the tables, but include the results nonetheless for transparency and ease of comparison.

**Limitations and Future Work** A notable exception to the observed trend of improved selective uncertainty occurs when making stable confidence predictions on simpler questions (e.g. average question lengths of CSQA and MMLU Conceptual Physics are less than half of others). We hypothesize that when questions resemble classical inductive classification tasks, the advantage of our test-time computation is less evident. Additionally, our analysis is limited in scope to multiple

choice datasets, leaving open-ended responses to future work. While entailment probability does help discount some logically incorrect explanations (Figure 1), there are still instances where it fails to properly distinguish. We test some alternatives to explanation faithfulness in Appendix B.5, but further exploration is needed. Efficiently sampling *high quality* explanations remains an open question as well. Our method adjusts the given explanation distribution based on plausibility, but better explanations may still exist that are not sampled by the LLM. One possible solution could involve using our entailment probability measure as a way to accept or reject incoming samples, increasing complexity but ensuring higher quality.

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

# APPENDIX

## A  EVALUATION OF UNCERTAINTY METRICS

In this section we provide formal definitions for each of the confidence evaluation metrics used. Consider the paired dataset $(x_i, y_i) \in \mathcal{D}$ where each datapoint $x_i$ has associated label $y_i$. Each $y_i$ takes on one value in the discrete set $\mathcal{Y} := \{1, 2, \dots, \ell\}$. Now our chosen prediction model $\phi$ outputs a prediction $\hat{y}_i := \phi(x_i)$ and our confidence function $f$ produces a score $f(x_i, \hat{y}_i) = r_i \in [0, 1]$. We use the indicator variable $c_i$ to denote whether the prediction is correct ($c_i := \mathbf{1}(y_i = \hat{y}_i)$). Lastly we define the full sequence of predictions $\hat{Y}$ and confidence predictions $R$ on dataset $\mathcal{D}$ of size $N$ as

$$\hat{Y} := \{\hat{y}_i = \phi(x_i) \mid x_i \in \mathcal{D}\} \tag{7}$$

$$R := \{r_i = f(x_i, \phi(x_i)) \mid x_i \in \mathcal{D}\} \tag{8}$$

**Expected Calibration Error (ECE)**  To calculate expected calibration error, we first group our data into $M$ partitions based on confidence interval. We denote the set of indices in each partition as:

$$B_m := \left\{ i \mid i \in N, \ \frac{(m-1)}{M} < r_i \leq \frac{m}{M} \right\} \tag{9}$$

Next, the empirical accuracy and average confidence functions for each partition are defined as

$$Acc(B_m) := \frac{1}{|B_m|} \sum_{i \in B_m} c_i, \quad Conf(B_m) := \frac{1}{|B_m|} \sum_{i \in B_m} r_i \tag{10}$$

Then the ECE is defined as the following weighted average:

$$\text{ECE}(R, \hat{Y}, M) := \sum_{m \in M} \frac{|B_m|}{M} |Acc(B_m) - Conf(B_m)| \tag{11}$$

The lower this error is, the better calibrated the model should be (with respect to the data distribution). While an easy metric to compute, there is a dependence on hyperparameter $M$ and in some cases variance within a partition with few samples will be high. To reduce this issue, we folllow (Nixon et al., 2019) in selecting adaptive partitions such that the number of *samples* are equal. That is our partitions are instead defined as

$$B_m' := \{i \mid i \in N, (m-1) * \lfloor N/(M-1) \rfloor < i \leq m * \lfloor N/(M-1) \rfloor \}. \tag{12}$$

Another well known issue with ECE is that when accuracy is very high, simply giving a high constant confidence estimate will result in very low calibration error (Ding et al., 2020; Xiong et al., 2023). Despite these drawbacks, we still choose to report the ECE metric as it is intuitive and serves as a common reference point with previous work.

**Area Under the Risk-Coverage Curve (AURC)**  For now, assume that $r_i \neq r_j \ \forall i \neq j$. Define the subset $R_{\geq r_i}$ as

$$R_{\geq r_i} := \{r \in R \mid r \geq r_i\} \tag{13}$$

We now say that the ordering map $\sigma : \{1, \dots, N\} \to \{1, \dots, N\}$ is the function that returns the dataset index $i$ of the $k$th largest element in $R$. Formally:

$$\sigma(k) := i \quad s.t. \ |R_{\geq r_i}| = k \tag{14}$$

To summarize so far, this ordering essentially gives us the dataset index of the $k$th most confident prediction. We can now finally define subsets of our most confident predictions as

$$\hat{Y}_K := \{\hat{y}_{\sigma(k)} \mid k \in \{1, \dots, K\}\} \tag{15}$$

The risk-coverage curve will measure the tradeoff between the size of $\hat{Y}_K$ and the accuracy. For each coverage level $h := K/N \in [0, 1]$, we plot the accuracy $Acc(\hat{Y}_K) \in [0, 1]$ to obtain the curve. Naturally $h = 1 \implies K = N$ and so the loss is simply the average model accuracy for the entire dataset. If our confidence measure is a good one, we expect higher accuracy when restricting our evaluation to a smaller subset of the most confident answers. Formally, the area under the risc-coverage curve (AURC) is is

$$\text{AURC}(R, \hat{Y}) := \sum_{K=1}^{N} Acc(\hat{Y}_K) \frac{K}{N} \tag{16}$$

**Area Under the Receiver Operator Curve (AUROC)**   For any binary classification problem, the receiver operator curve looks at the tradeoff between false positive rate $\alpha$ (plotted on the x-axis) and true positive rate $\beta$ (y-axis), based on retaining only predictions with scores above some threshold $t$. We denote a thresholded set of predictions as $\hat{Y}_t := \{y_i \in \mathcal{D} \mid r_i > t\}$, and $t_\alpha$ as the threshold such that $\text{FP}(\hat{Y}_{t_\alpha}) = \alpha$. If we have built a perfect classifier of correct and incorrect predictions, there should exist a threshold $t_0$ for which $\hat{Y}_{t_0}$ contains all of the predictions the model got right and none of which it got wrong. This would correspond to a true positive rate of $\beta = 1.0$ for all false positive levels $\alpha \in [0, 1]$. Conversely, if confidence metrics were generated at random, any $X_t$ is likely to contain just as many false positives and true positives, and so the ROC curve will resemble a diagonal line. Therefore we would like the area under the reciever operator curve to be as closer to 1 as possible. Formally, this area is written as

$$\text{AUROC}(R, \hat{Y}) := \int_0^1 \text{TP}(\hat{Y}_{t_\alpha}) d\alpha, \tag{17}$$

# B   EXPERIMENTAL DETAILS

In this section we discuss the implementation details of LLM prompts, dataset characteristics, and evaluation methods. We also include additional experiments examining the effect of explanation hyperparameters.

## B.1   PROMPTS

In this section we provide the prompts used for each confidence elicitation method. Text in red represents substitutions that are made to the prompt at inference time, for example adding the text of the specific multiple choice question. For the **stable explanations** method in Figure 3 we provide our explanation generation prompt and conditional answer generation prompt. We use the response from this first prompt to generate our default question explanations (discarding the answer that comes after). We then use the logits from the second prompt conditioned on explanations as the posterior answer distribution for that explanation. The entailment probability prompt used is the same as in (Sanyal et al., 2024). For the **token probability** prompt (Figure 4) we use a simple question and answer format, and use the softmax of next token logits to determine answer confidence. For the **linguistic confidence** prompt in Figure 5 we follow (Shrivastava et al., 2023) best prompt choice and parse the returned response for answer and confidence value. For **chain-of-thought consistency** confidence we use a zero-shot modified version of the prompt from (Fu et al., 2023) (Figure 6) to generate multiple explanations and answers (discarding explanations and taking a majority vote over returned answers). We also explore using this prompt to generate explanations (discarding answers instead) for our CoT-stability confidence metric. The **top-k** confidence prompt is provided in Figure 7; the resulting LLM response is parsed for $k$ confidence values. Lastly we include the conditional explanation prompt used to generate correct and incorrect explanations in Figure 1. Unless otherwise noted, temperature for all generated explanations is set to Temp=0.7 for both stable explanations and CoT-consistency method.

```
Token Prob Confidence Prompt:
Question: [multiple choice question]
Answer:
```

Figure 4: Token Probability Prompt

**Stability Explanation Prompt:**
Read the given question and select the most appropriate
answer by indicating the associated letter. Please output
strictly following the explanation-then-answer format:
Explanation: <detailed reasoning steps> Answer: (letter)

Question: [multiple choice question]

**Stability Conditional Answer Prompt:**
You are an expert analyst considering arguments from
different perspectives. Given a question and an argument,
choose the correct answer. You must answer the question
with one of the valid choices. You must provide only a
single answer.
Argument: Given the scenario in the question, [explanation]
Answer: The correct answer is

**Entailment Prompt:**
Premise:[multiple choice question]
Hypothesis:[explanation]
Question: Given the premise, is the hypothesis correct?
Answer (T/F):

Figure 3: Stable Explanation Prompts

**Linguistic Confidence Prompt:**
Answer the following question to the best of your ability, and provide a
score between 0 and 1 to indicate the confidence you have in your answer.
Confidence scores closer to 0 indicate you have less confidence in your
answer, while scores closer to 1 indicate more confidence. You must answer
the question with one of the valid choices. You must provide only a single
answer.

Question: This is a question
(A) first answer
(B) second answer
(C) third answer
(D) fourth answer
(E) fifth Answer
Answer: (D)
Confidence: 0.4

Question: This is another Question
(A) first answer
(B) second answer
(C) third answer
(D) fourth answer
(E) fifth Answer
Answer: (A)
Confidence: 0.7

Question: [multiple choice question]

Figure 5: Linguistic Confidence Prompt

```
CoT Explanation Prompt:
You will be given a question at the end, after the examples, for which you
are to select the most appropriate answer by indicating the associated
letter. Please first output step-by-step reasoning about how to solve the
question. Then output the answer. You MUST output exactly one of the provided
answers.

Q: This is a question
Which one of the choices is correct, (A), (B), (C) or (D)?
Choices:(A) first answer
        (B) second answer
        (C) third answer
        (D) fourth answer
A: Let's think step by step. Given the scenario, we know that answer cannot
be (B) or (C) because... From here we continue our line of reasoning...
Therefore, the answer is (A).

Q: This is another question
Which one of the choices is correct, (A), (B), (C) or (D)?
Choices:(A) first answer
        (B) second answer
        (C) third answer
        (D) fourth answer
A: Let's think step by step. This is more step-by-step reasoning
Therefore the answer is (C).

Q: [multiple choice question]
A: Let's think step by step.
```

Figure 6: Chain of Thought Explanation Prompt

```
Top-K Confidence Prompt:
The task is to read the given question and select the most appropriate
answer by indicating the associated letter. Provide your {k} best guesses
and the probability that each is correct (0.0 to 1.0) for the following
question. Give ONLY the guesses and probabilities, no other words or
explanation.

For example:
G1: <first most likely guess, as short as possible; not a complete
sentence, just the guess!>
P1: <the probability between 0.0 and 1.0 that G1 is correct, without any
extra commentary whatsoever; just the probability!>
...
GN: <Nth most likely guess, as short as possible; not a complete sentence,
just the guess!>
PN: <the probability between 0.0 and 1.0 that GN is correct, without any
extra commentary whatsoever; just the probability!>

Question: [multiple choice question]
```

Figure 7: Top-K Confidence Prompt

## B.2 DATASET DETAILS

We can observe in Appendix B.2 that the QA datasets with longer questions typically are harder for the model to answer correctly. We see that our method, like many other sample+aggregate based answering methods generally has higher accuracy than the baseline model (Wang et al., 2022). This accuracy boost is less pronounced however for GPT-4.

For GPT-3.5-Turbo results we generate confidence scores for 500 questions per dataset (or maximum dataset size if smaller). Due to computational cost we only use 200 questions per dataset when

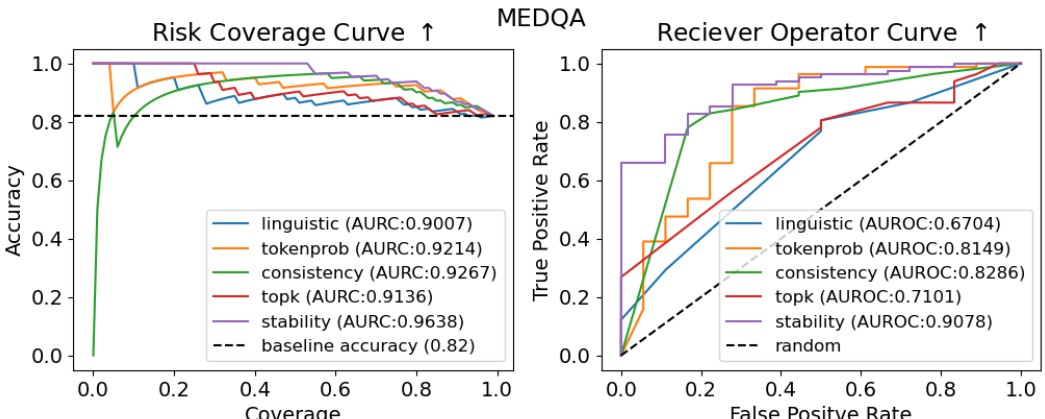

Figure 8: Risk coverage (left) and receiver-operator (right) curves for confidence metrics generated on the MedQA questions using GPT-4. Our stability method outperforms others on this dataset as evidenced by larger area under the curves. We can also observe that questions with confidences in the top 50% were all correct.

testing on GPT-4-Turbo. We use validation splits for CSQA, TruthQA, and test splits for MedQA and MMLU datasets.

| Method | Avg. Question Length | GPT-3.5 Acc. | GPT-3.5 Stability Acc. | GPT-4 Acc. | GPT-4 Stability Acc. |
|---|---|---|---|---|---|
| CSQA | 151 | 0.79 | 0.80 | 0.84 | 0.88 |
| TruthQA | 329 | 0.54 | 0.64 | 0.85 | 0.91 |
| MedQA | 916 | 0.59 | 0.68 | 0.82 | 0.84 |
| MMLU Law | 1233 | 0.46 | 0.56 | 0.64 | 0.67 |
| MMLU Physics | 172 | 0.57 | 0.72 | 0.92 | 0.92 |

Table 3: Comparing accuracy for default model predictions vs. most confident stability predictions across benchmark datasets. One can observe a clear improvement in accuracy for both GPT-3.5 and GPT-4.

### B.3 EVALUATION DETAILS

When evaluating confidence methods, it is important to note that performance implicitly depends on the prediction set $\hat{Y}$. For example, a metric may be well calibrated on correct answers but still be overconfident on incorrect ones, meaning the confidence metric would evaluate as worse on a less accurate prediction set. Therefore, for comparison purposes we use the same set of default LLM predictions (setting Temp=0) for GPT-3.5 and GPT-4 results.

In order to break possible ties in confidence when evaluating AURC and AUROC methods, we follow the approach of (Shrivastava et al., 2023) and add a small amount of gaussian noise ($\sigma = 1e - 6$) to each confidence score. We repeat this process for $r = 10$ times and take the average AURC and AUROC scores. We also follow common practice in previous works by using $M = 10$ as the number of bins when calculating ECE (Achiam et al., 2023).

We use OpenAI's *gpt-3.5-turbo-1106* snapshot for GPT-3.5 experiments and *gpt-4-1106-preview* snapshot for GPT-4. Generating and evaluating confidence scores for each method on one dataset takes on the order of an hour for GPT-3.5-Turbo, and two hours for GPT-4 using OpenAI's API.

### B.4 EFFECT OF EXPLANATIONS ON ANSWER ENTROPY

We compare the entropy of the default model answer distribution $p_\phi(a|q)$ to the entropy after conditioning on a CoT-generated explanation $p_\phi(a|e, q)$ (e.g. using the prompt from figure 6). In

Figure 9 we find that for the majority of questions (>75%), the entropy becomes smaller as the model becomes more confident in a single answer.

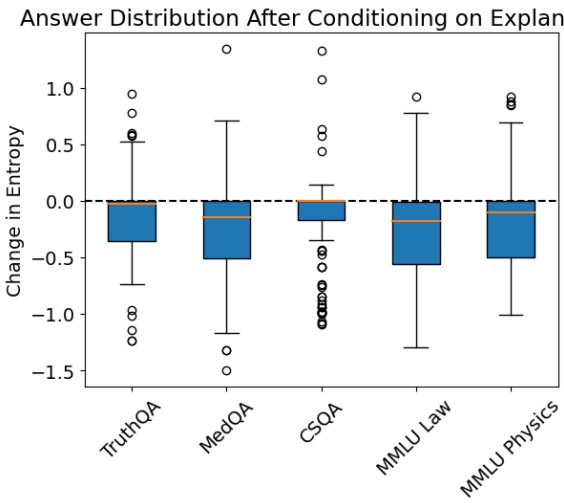

Figure 9: Difference in entropy the of answer distribution before and after conditioning on a CoT-style explanation for GPT-3.5.

## B.5 ALTERNATE EXPLANATION PLAUSIBILITY MEASURES

Inspired by (Kadavath et al., 2022), which looks at the true/false token probability an LLM assigns to a given answer being true, we explore evaluating the probability that an explanation is 'true'. To do this, we provide the model with both question and explanation and ask: 'Is this the most likely explanation? (T/F)'. We also try asking the question 'Does the explanation completely describe the question? (T/F)'. We then repeat the experiment in Section 4.2, examining distributions of explanations measured via these probabilities. We find in figure 10 that these measures fail to properly distinguish between different explanations.

## B.6 SENSITIVITY TO EXPLANATION PROMPTING

Our stable explanation method reweights explanations based on entailment probability, but if the quality of sampled explanations is poor to begin with our resulting distribution will still be inaccurate. Here we will discuss the effect of instructing the LLM to generate explanations before or after an answer (i.e. the order of 'explanation' and 'answer' in the stability explanation prompt in Figure 3). We observe in Appendix B.6 that generating explanations before the answer clearly results in higher quality explanations, as evidenced by improved performance on selective uncertainty and calibration tasks.

| | Pre-Answer Stability (Default) | | | Post-Answer Stability | | |
|---|---|---|---|---|---|---|
| | AURC ↑ | AUROC ↑ | ECE ↓ | AURC ↑ | AUROC ↑ | ECE ↓ |
| *CSQA* | 0.901 | 0.779 | 0.123 | 0.866 | 0.731 | 0.201 |
| *TruthQA* | 0.801 | 0.853 | 0.21 | 0.792 | 0.839 | 0.254 |
| *MedQA* | 0.784 | 0.798 | 0.169 | 0.743 | 0.743 | 0.251 |
| *MMLU Law* | 0.642 | 0.736 | 0.259 | 0.629 | 0.706 | 0.289 |
| *MMLU Physics* | 0.792 | 0.834 | 0.186 | 0.779 | 0.811 | 0.252 |

Table 4: Comparing stability confidence performance using explanations generated before and after an answer for GPT-3.5. One can clearly observe that explanations generated before the answer (i.e. in chain-of-thought fashion) outperform those generated afterwards across all performance metrics.

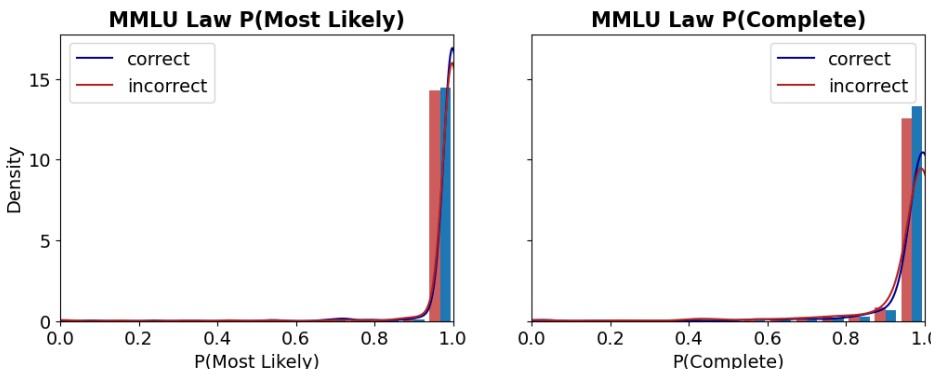

Figure 10: Empirical distribution of MMLU explanations when measured via GPT-3.5 probability of being 'most-likely explanation' (left) and probability of 'completely describing' the question (right). One can see that true (blue) and false (red) answer-conditioned explanations are difficult to distinguish.

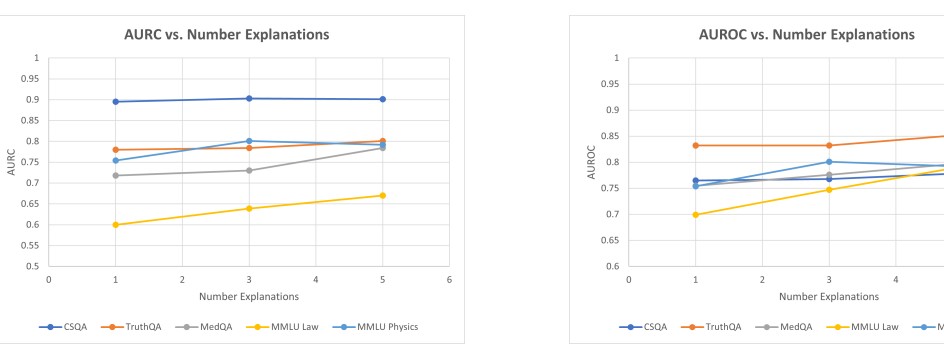

(a) AURC vs. Number of Explanations for the stable explanations confidence metric.

(b) AUROC vs. Number of Explanations for the stable explanations confidence metric.

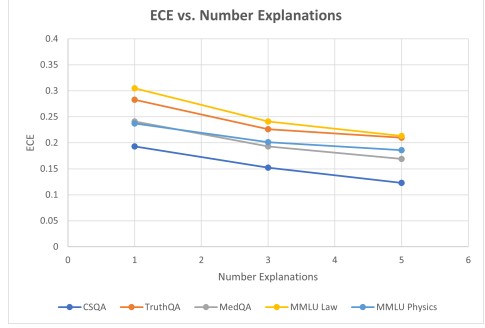

(c) ECE vs. Number of Explanations for the stable explanations confidence metric.

Figure 11: Comparison of stable explanation confidence using different numbers of explanations per question ($M = \{1, 3, 5\}$). Testing is done on GPT-3.5-Turbo for all five benchmark datasets. One can observe improving but saturating performance for each dataset.

## B.7 VARYING SAMPLE SIZE

In this section we briefly analyze the effect that the number of sampled explanation has on our confidence metric. In Figure 11 we observe that selective uncertainty performance (AURC and AUROC) saturates quickly for simpler questions answering tasks such as commonsenseqa. On the other hand MedQA and MMLU Law datasets both demonstrate steady performance gains up to $M = 5$ samples per question. Calibration error gradually decreases for all datasets examined.

## B.8 COMPARISON TO SEMANTIC ENTROPY

Semantic Entropy (Kuhn et al., 2023) reduces to the naive entropy in the case of multiple choice questions, as answer clusters are already well defined (i.e. no variation syntactically). Therefore we have chosen to use answer token probability as our main confidence metric baseline. However, we have added additional experiments demonstrating that directly using the entropy metric corresponds to the token probability almost exactly when it comes to the selective uncertainty task. This is unsurprising as the model typically assigns most probability to a single token, meaning the entropy is strongly dependent on this specific confidence.

| | Method | CSQA | TruthQA | MedQA | MMLU Law | MMLU Physics | Average |
|---|---|---|---|---|---|---|---|
| AURC ↑ | Token Prob. | 0.914 | 0.75 | 0.727 | 0.622 | 0.751 | 0.753 |
| | Semantic Entropy | 0.912 | 0.747 | 0.722 | 0.62 | 0.752 | 0.751 |
| AUROC ↑ | Token Prob. | 0.806 | 0.772 | 0.702 | 0.676 | 0.747 | 0.741 |
| | Semantic Entropy | 0.808 | 0.775 | 0.701 | 0.673 | 0.75 | 0.741 |

Table 5: Comparison of semantic entropy metric with the token probability metric on GPT-3.5. Results for AURC and AUROC are almost identical.

| | Method | CSQA | TruthQA | MedQA | MMLU Law | MMLU Physics | Average |
|---|---|---|---|---|---|---|---|
| AURC ↑ | Token Prob. | 0.941 | 0.939 | 0.91 | 0.828 | 0.936 | 0.911 |
| | Semantic Entropy | 0.94 | 0.94 | 0.917 | 0.825 | 0.939 | 0.912 |
| AUROC ↑ | Token Prob. | 0.798 | 0.842 | 0.784 | 0.745 | 0.813 | 0.796 |
| | Semantic Entropy | 0.799 | 0.842 | 0.787 | 0.741 | 0.803 | 0.794 |

Table 6: Comparison of semantic entropy metric with the token probability metric on GPT-4. Results for AURC and AUROC are almost identical.

## B.9 COMPARISON TO TTA

Contemporaneously to this manuscript's submission, another method related to our approach was proposed (Li et al., 2024). The Think-Twice before assure (TTA) method asks for explanations conditioned on different answers, then does a top-k confidence elicitation using these explanations in the prompt. Although similar in the sense that confidence metrics are being generated by conditioning on explanations, their combination of explanations into a single prompt does not match the ensemble of test-time classifiers view that our method takes. The authors have not yet released code or dataset splits, but we have implemented their method by following the written procedure and using the same prompts (see Figure 12). We found during our implementation on the shared CSQA dataset, the evaluation results for selective uncertainty tasks are slightly below that what the authors report (AURC, AUROC), most likely due to the difference in specific questions used during testing. Nonetheless we report the full results of our implementation in table 7, and note that this metric does appear to have lower ECE in many cases.

| | TTA (Our Implementation) | | | |
|---|---|---|---|---|
| | AURC ↑ | AUROC ↑ | ECE ↓ | Acc. ↑ |
| CSQA | 0.885 | 0.688 | 0.104* | 0.736 |
| TruthQA | 0.698 | 0.706 | 0.093* | 0.672* |
| MedQA | 0.641 | 0.581 | 0.207 | 0.505 |
| MMLU Law | 0.574 | 0.657 | 0.148* | 0.456 |
| MMLU Physics | 0.717 | 0.697 | 0.1* | 0.557 |

Table 7: Evaluation for the TTA Confidence metric (Our implementation) on GPT-3.5. Results that outperform our stable explanations metric are marked with an asterisk.

```
TTA Explanation Prompt:
The task is to read the given question and select the most appropriate
answer by indicating the associated letter.

Question: [multiple choice question]
Answer: [answer text]

Please generate an explanation to try to justify the answer judgement.

TTA Confidence Prompt:
[Top-K Prompt]
Possible explanation 1: [explanation 1]
Possible explanation 2: [explanation 2]
…
Possible explanation N: [explanation N]
```

Figure 12: TTA Confidence Prompt

## C  ALTERNATIVE PERSPECTIVES OF STABLE EXPLANATIONS

### C.1  CONFIDENCE THROUGH THE VIEWPOINT OF TRANSDUCTIVE INFERENCE

Transductive learning selects a classifier at inference-time based on a combination of training and test data (Dhillon et al., 2020; Vapnik; Joachims et al., 1999). Typically transductive learning involves fine-tuning some classifier parameter $w$ based on an explicit loss objective. However, we claim that using an LLM to generate a sequence of text before an answer (i.e. an explanation) is an alternate way of doing transductive reasoning. First, note that answering a question after an explanation, such as in chain-of-thought prompting (Wei et al., 2022), effectively changes the decision boundary of the LLM classifier at inference time. Second, consider that when an LLM generates an explanation, it produces concepts related to those in the question. These additional concepts can be thought of as forcing the LLM at inference time to pay more attention to the decision boundary in the area around the test datum. In-context learning literature, which examines LLM performance after *manually* inserting demonstrations similar to the test question, has already shown a direct connection between transformer context adjustment and classical fine-tuning behavior (Dai et al., 2022).

To formalize this perspective, let $D^t = \{(x_1, y_1), \ldots, (x_t, y_t)\}$ be a dataset of sequential data up to time $t$, with $x_i \in X \subset \mathbb{R}^M$ and labels $y_i \in Y \subset \{1, \ldots, K\}$. We denote with $D^t_- = \{(x_1, y_1), \ldots, (x_{t-1}, y_{t-1}), x_t\}$ the dataset without the last label $y_t$. We can write our transductive prediction for $x_t$ given data $D^t_-$ *including* $x_t$ as:

$$\hat{y}_t = \underset{w,y}{\operatorname{argmin}} \underbrace{\frac{1}{t}\ell(f_w(x_t), y) + \frac{1}{t}\sum_{i=1}^{t-1}\ell(f_w(x_i), y_i)}_{\doteq L(w;(D^t_-,y))}. \tag{18}$$

If $\ell$ is interpreted as a log likelihood, then $L$ can be interpreted as the negative log posterior probability over hypotheses. If we think of optimizing instead over explanations where $f_e(x_t) = \phi(x_t, e)$, then the problem reduces to finding an explanation that strongly supports a single answer without biasing predictions on original test data. The second term in equation (18) is expensive to compute at inference time, but if some approximation of this training loss existed it would make optimization tractable. We hypothesize that if the explanation under consideration is plausible and faithful to the question (as determined using the same LLM), it should not reduce the accuracy of previous decisions too much. Therefore we can avoid having to optimize over all previous questions and instead optimize over whatever faithfulness measure $g_\phi(e)$ we define:

$$\hat{y}_t = \underset{e,y}{\operatorname{argmin}} \, \ell(\phi(x_t, e), y) + \lambda g_\phi(e) \tag{19}$$

This looks exactly like the typical transductive setting but with a more easily computable 'transductive prior'.

## C.2 Confidence Throught the Viewpoint of Solomonoff Induction

While transductive inference typically finds single test-time classifier, our method looks for a *distribution* of likely classifiers. In this sense, our method can be seen as a special case of Solomonoff induction (Kolmogorov, 1965). Solomonoff induction considers how well data-generating programs, $H$, (i.e. a binary string run on a Turing machine) explain the test data, $D$

$$P(H|D) = \frac{P(D|H)P(H)}{P(D|H)P(H) + \sum_{A \neq H} P(D|A)P(A)}, \tag{20}$$

where $A$ are alternative programs. Solomonoff induction formalizes the principle of Occam's razor by choosing a universal prior $P(H)$ that gives a higher probability to shorter-length programs. Then to predict new data $D'$ given previous observations, one simply computes

$$P(D'|D) = \mathbb{E}_H[P(D'|H, D)] = \sum_H P(D'|H, D)P(H|D). \tag{21}$$

While these Bayesian equations seem simple, Solomonoff's induction is provably uncomputable. However, our method can be interpreted as restricting our hypothesis class from the set of all computable programs $H$ to the set of all *LLM-interpretable* programs $e$. Instead of a prior on program length, we can use the LLM's prior likelihood of valid sequences in the language $p_\phi(e)$. This restriction makes our calculations more tractable, as we can easily approximate expectations over our hypothesis class by sampling explanations from the LLM.

## C.3 Confidence Through the Viewpoint of Stability

Another recent line of work has been analyzing LLMs through the lens of stochastic dynamical models (Soatto et al., 2023). Through the perspective of stability analysis one could interpret our method's preference for explanations convening to a single answer as searching for *fixed points* of a specific LLM system. This LLM dynamical system consists of two alternating steps, first generating an explanation conditioned on one of the answers ($e \leftarrow \phi(q, a)$) then generating a new answer based on this explanation ($a' \leftarrow \phi(q, e)$). Intuitively this system mirrors how a human expert may think about a question by considering alternative conclusions one could draw given beliefs about the world. An answer with only a single plausible explanation that strongly supports that same answer (i.e. decision distribution collapses to a singleton) forms a stable cycle in this system.

