# OpenReview forum: "Measuring LLM Confidence through Stable Explanations"
_ICLR.cc/2025/Conference — Submitted to ICLR 2025_

### Official Review · Reviewer_b15e · 2024-10-22

**Soundness:** 2
**Presentation:** 3
**Contribution:** 2
**Rating:** 5
**Confidence:** 4

**Summary:**

The authors propose to estimate the uncertainty of LLM answers in single-token QA tasks by not directly recording an answer (probability) after giving a question, but asking for an explanation in-between. This is resampled multiple times per question and reweighted by how much the explanation fits to the question, judged by an NLI algorithm. First results show that this might improve the correctness AUROC on some datasets and models.

**Strengths:**

* The authors use five datasets that are challenging to current LLMs, as opposed to simple datasets like TriviaQA.
* Chain of thoughts / explanation prompting is a topic with increased current interest.
* The limitation section is well-written, and I would love to see more experimental analyses mentioned there (such as the difference in performance on simple / hard questions)

**Weaknesses:**

EDIT: Post-rebuttal, I've increased my score from 3 to 5 due to the new experiments and variances reported. Below are my weaknesses for the original score.

* Many common baselines are missing, even if they are cited and discussed in the paper. I would have a higher trust in the results (and increase my soundness score) if the paper compared to P(True), Semantic Entropy, the entropy of the output distribution (eq. to Naive Entropy, because we only have one token), the number of different answers when prompted multiple times, and the Eccentricity of the NLI entailment graph between answers. To get you started quickly, I recommend the codebase of https://arxiv.org/abs/2311.07383 (I'm not author or affiliated with any authors of that paper, it's what I genuinely believe will help you the best and the fastest)
* The TTA approach has a higher ECE than the proposed method on 4 of the 5 datasets, but is hidden into Appendix B.7 because it is contemporary work. I agree that the underperformance should not be a reason for rejection because of the comtemporary nature of the work (and also because I don't believe that papers should be published if and only if SOTA) and it does not influence any of my scores (which I made before discovering it in B.7), but I strongly recommend bringing the results to the main paper for scientific integrity.
* The authors only use 100-250 questions per dataset (Appendix B.2). This is likely to have large variance, but the variance is not reported. It could be that it is in the region of the recorded AUROC differences (~0.01 to 0.05). This lowers my soundness score. I believe a higher number of questions would be an easy way to improve the robustness of the experiments, since the experiments per Appendix B.3 currently take only 15 GPU hours (or more precisely, OpenAI-API hours).
* Section 4.2 asks an interesting side-question, but makes an experimental choice which I believe confounds the results. That is, the (sequence) probability of explanations of wrong and correct answers is compared _after_ conditioning the generation of the explanation on the answer. Due to the autoregressive nature of Transformers, this will naturally lead to similar probabilities. Do you have data for explanations generated before giving the answer, i.e., the sequence probability of the "middle part" of a "question + explanation + answer" sequence (which is more aligned with what your method does)? This would be a highly interesting experiment, and allow to increase my soundness score.
* The approach is limited to QA tasks with a single output token. This is a very synthetic setting compared to free-form answers, which limits the contribution score.
* Experiments are limited to GPT-3.5 and GPT-4. These are very similar LLMs. It would be interesting to see whether the results transfer to, e.g., Llama-3.1-8b-chat (or Mistral/Falcon/Yi).
* Code and data are not released, hindering the replication of results.
* The notation is not clearly defined, hindering exact replication at the current state of the paper and being the prime reason for my presentation score (because the writing itself is good!). I.e., the (set?) $E$ is never defined. $\mathcal{A}^\tau$ is never defined. $a$ and $i$ can probably be used interchangably, since $a$ is always only a single token, bijectively identifiable by its index $i$. It's not defined whether the softmax is over the alphabet $\mathcal{A}$ or the set of possible answers $S$, and how that set $S$ is defined in the first place.
* The paper repeatedly makes the argument that the underperformance of token-likelihood based uncertainty estimates is due to a train-test distribution shift, particularly in the introduction and related works sections. However, there is no experimental data to back this claim and no data to show that the proposed method is better in this regard, such as transfer experiments or in-distribution vs out-of-train-distribution experiments.
* The approach requires to sample multiple explanations, i.e., multiple sequences of additional tokens. This makes it expensive.

Minor comments that do not influence my rating and do not need to be addressed in the rebuttal (only in the final camera ready):
* I found Appendix C (mathematical perspectives on your work and other works) much more interesting than Section 2.1 (general intro into uncertainty in ML). I think swapping the two would make the paper more interesting to expert readers.
*  "is called well-calibrated if on average predictions with confidence r = 0.XX are correct close to XX% of the time." could be rewritten to something like "... $r \in [0,1]$ are correct $100r \\%$ of the time."
* typo in line 198: sef-consistency
* I think Figure 2 should be a Table
* Personal take, but I would tone done the transductive learning narrative, which only really gets explained in the conclusion. I believe this paper falls more into the category of CoT approaches, which is a great topic in and by itself.

**Questions:**

* Do you generate the explanation before or after generating the answer? From the text, I understood that you use "Question, Explanation, Answer", but in Figures 1 (and 8 in the appendix) it looks the other way around. Is this only for the experiments in Section 4.2?
* Why did you set the temperature to 0.7 as opposed to 1? Do you have data supporting this choice? (E.g., higher accuracy)
* Can you report the missing baseline approaches, other models, variance estimates, and potentially higher number of questions per dataset, as outlined above?
* Can you add the updated section 4.2 experiment, as highlighted above?

---

> ### Comment · Reviewer_b15e · 2024-11-25
>
> Dear authors,
> if you happen to have new data on the baselines, dataset sizes, or the order of the explanations, I'd be more than happy to discuss the ratings, as I've indicated in the review. I do believe that your general topic, combining uncertainty and explanations, is promising.

---

> ### Author Response · Authors · 2024-11-28
> **Response to Reviewer b15e**
>
> We sincerely thank the reviewer for the constructive comments and thoughtful suggestions. Please see the overall comment for concerns regarding **increasing number of samples**, the **explanation conditioning experiment**, and **open-ended generation**. We aim to address your specific concerns as follows:
>
> **Many common baselines are missing:**  Thank you for the additional baseline suggestions.
> * We have added the **P(true)** confidence metric from Kadavath et al. 2022 to the main results (see overall comment).
> * **Semantic Entropy** (Kuhn et al., 2023) reduces to the naive entropy in the case of multiple choice questions, as answer clusters are already well defined (i.e. no variation syntactically). While we therefore use answer token probability directly as our main confidence metric baseline, we have added additional experiments (Appendix B.7 of the revised manuscript) demonstrating that using the entropy metric corresponds to the token probability almost exactly when it comes to the selective uncertainty task. This is unsurprising as the model typically assigns most probability to a single token, meaning the overall entropy is strongly dependent on this specific confidence.
> * For a similar reason we chose not to include uncertainty quantification arising from different calculations made from the answer similarity matrix (Lin et. al 2024). In the case of multiple choice answers this graph Laplacian becomes block diagonal, and the **NLI eccentricity** confidence estimate becomes similar to counting the frequency of responses (except now using an l2 norm). However, we agree that these semantic similarity baselines should be included for future work on open-ended generation (see further discussion in overall comment). We also appreciate the codebase suggestion and certainly will use this to help standardize future experiments!
>
> **Using a higher number of samples:** We have doubled the number of samples for each experiment, resulting in similar relative improvements in selective uncertainty (see overall comment).
>
> **Release of Code:** We will release our code repo in the camera-ready submission.
>
> **Notational Clarification:** Thank you for the feedback on notation, we have updated definitions for the sampled explanation set $E$, and $\tau$-length token sequence $\mathcal{A}^\tau$. We clarify that the softmax is taken over the answer subset $\mathcal{S}$, and this set is determined by the multiple choice question itself (not necessarily fixed between questions).
>
> **Sampling multiple explanations is expensive:** While we agree that sampling multiple explanations can be expensive, this is on par with methods like CoT consistency and Semantic Entropy (Wang et al. 2023,Kuhn et al. 2023). Additionally, one could imagine using a smaller version of the model for efficient explanation generation while using the larger model to verify logical consistency and compute conditional answer distributions (similar in spirit to speculative decoding )(Leviathan et al 2023).
>
> **Choice of Temperature:**  We follow the same temperature setting as suggested in the CoT consistency paper (Wang et al 2023, Figure 4), which also shows robustness to temperature variation in the range of $T\in (0.5,0.95)$.
>
> **Exploring Additional Models, Out-of-Distribution Performance:** Please see overall comment
>
> ---
> [1] Kadavath, Saurav, et al. "Language models (mostly) know what they know." arXiv preprint arXiv:2207.05221 (2022).
>
> [2] Kuhn, Lorenz, Yarin Gal, and Sebastian Farquhar. "Semantic uncertainty: Linguistic invariances for uncertainty estimation in natural language generation." arXiv preprint arXiv:2302.09664 (2023).
>
> [3] Lin, Zhen, Shubhendu Trivedi, and Jimeng Sun. "Generating with confidence: Uncertainty quantification for black-box large language models." arXiv preprint arXiv:2305.19187 (2023).
>
> [4] Wang, Xuezhi, et al. "Self-consistency improves chain of thought reasoning in language models." arXiv preprint arXiv:2203.11171 (2022).
>
> [5] Leviathan, Yaniv, Matan Kalman, and Yossi Matias. "Fast inference from transformers via speculative decoding." International Conference on Machine Learning. PMLR, 2023.

---

> > ### Comment · Reviewer_b15e · 2024-11-28
> >
> > Thank you for the answer and for providing additional experiments!
> >
> > > We have also reworded the mentions in introduction of transductive inference, providing a clearer definition/reference and emphasizing the balance between learning model weights during training and reasoning chains during inference.
> >
> > I've read the updated introduction, and though I understand your intuition that explanations help sharpen the generalization boundary, I would refrain from claiming this because there are no experiments that support this. All experiments are in-distribution, so there are no generalization tests. I believe this claim is not required for the paper.
> >
> > > OOD Generalization (general response)
> >
> > I appreciate your thoughts and the qualitative example. However, as my intial review and that of reviewer dNm6 noted, for such a strong statement (why and when your approach works), theoretical or quantitative support would be required. But as I was also mentioning, your paper does not hinge on this connection to transductive learning in any way, so it would also be possible to just not bring it up, or mention it only as a speculation in the discussion section.
> >
> > > additional baseline suggestions
> >
> > Thank you. This increases my scientific soundness score.
> >
> > > We have doubled the number of samples for each experiment
> >
> > Thank you. Upon comparing the results from the 150 to the 300 samples, there seem to be some quite heavy differences. E.g., GPT4 AURC CoT-Stability MMLU Law goes from 0.78 to 0.85 and is now the best method, GPT4 AUROC CoT-Stability MMLU Physics goes from 0.88 to 0.95, GPT4 AUROC Linguistic TrustQA dropped from 0.75 to 0.7. Overall, many results changed between -0.07 and +0.07 between the submission with lower number of samples samples and the revision with higher number of samples (not taking into account the ECE changed as I understand you've changed the metric). These are quite drastic changes, and I'm not sure if they will not change again when using large enough sample sizes. Since variances are also not reported, I am still unclear as to which outperformances are actually stable and which can be explained by noise, as already mentioned in the initial review. The too low sample size still limits my soundness and overall score.
> >
> > > We will release our code repo in the camera-ready submission
> >
> > Thank you. That also contributes to increasing my soundness score.
> >
> > > While we agree that sampling multiple explanations can be expensive, this is on par with methods like CoT consistency and Semantic Entropy
> >
> > It is true that these methods are also expensive (though the comparison to Semantic Entropy is somewhat unfair, because it is only required to handle open-ended questions and reverts to more or less token probabilities in the single-token QA sets studied in this paper, as the authors confirmed). But this is also a known and often cited drawback of these methods, so the weakness of stable explanations is likely to face similar criticism in literature and by practitioners.
> >
> > > Choice of Temperature
> >
> > Thanks for clarifying this.
> >
> > > Additional Models
> >
> > I humbly disagree that further model architectures are not so important because they tend to scale similarly in other works. In the initial review I (and also reviewer dNm6) suggested more architectures, especially open-source ones, because 1) they are more reproducible, 2) Especially in your new topic of explanations, we have no guarantees the models behave similarly, and 3) As mentioned above, the current results are quite noisy, so it would be a means to obtain more data. This is why I still believe that only surveying GPT 3.5 and 4 is a weakness in terms of the contribution and soundness scores.
> >
> > > explanation conditioning experiment
> >
> > I understand that you have changed the example in Figure 1 to question -> explanation -> answer. I just wanted to ask if the results in Figure 2 (which I still believe should be labeled at a Table) are also with this updated form or remain with the question -> answer -> explanation variant?
> >
> > As of now, I am increasing my soundness score to 2 as promised and remain at your disposal.

---

> > > ### Author Response · Authors · 2024-12-02
> > > **Author Response**
> > >
> > > We appreciate your feedback on our revisions. Thank you for raising your soundness score, we hope you will consider raising your overall score as well in response to the additional variance information provided below.
> > >
> > > > Explanation conditioning experiment
> > >
> > > Our confidence method samples explanations without conditioning on an answer (question -> explanation -> answer) and we have not changed this in our revision. Therefore Figures 1 and 2 now exactly match in the way explanations are generated. Figure 2 will also become a table in the camera-ready version.
> > >
> > > > Experiment Sample Size
> > >
> > > While adding more samples certainly changes the raw AURC+AUROC metrics, we emphasize that the relative performance gains remain similar across revisions. We have also provided the table below with the AURC and AUROC variances (maximum across all methods tested) for each dataset and model. This is calculated via a standard bootstrapping procedure [1]. One can observe that even in the lower sample scenario with GPT-4, the variance is always much smaller (less than half) than the gap between our method and the next best, except for AUROC on MMLU Law (where both our method and CoT-Consistency are effectively tied).
> > >
> > > |               |                | CSQA    | TruthQA | MedQA   | MMLU Law | MMLU Physics |
> > > |---------------|----------------|---------|---------|---------|----------|--------------|
> > > | GPT-3.5-Turbo | AURC Variance  | <=0.002 | <=0.002 | <=0.002 | <=0.003  | <=0.003      |
> > > |               | AUROC Variance | <=0.003 | <=0.003 | <=0.002 | <=0.002  | <=0.002      |
> > > | GPT-4         | AURC Variance  | <=0.005 | <=0.001 | <=0.002 | <=0.003  | <=0.001      |
> > > |               | AUROC Variance | <=0.007 | <=0.004 | <=0.003 | <=0.005  | <=0.005      |
> > >
> > >
> > >
> > > > OOD Generalization, Transductive Terminology
> > >
> > > Lastly, we appreciate your perspectives on the transductive terminology and generalization claims and will consider them in the final version of our paper.
> > >
> > > ---
> > > [1] https://search.r-project.org/CRAN/refmans/pROC/html/var.html

---

### Official Review · Reviewer_4HSJ · 2024-11-04

**Soundness:** 3
**Presentation:** 2
**Contribution:** 3
**Rating:** 6
**Confidence:** 3

**Summary:**

* The paper proposes a novel framework for measuring LLM uncertainty by analyzing the distribution of generated explanations. The key insight is combining explanation sampling with a reweighting mechanism based on logical consistency, measured through entailment probability.
* The method is evaluated comprehensively across 5 datasets using both GPT-4 and GPT-3.5. Performance is primarily assessed using AURC and AUROC metrics, with several competitive baselines for comparison.

**Strengths:**

* Strong empirical results with thorough experimental validation.
* Well-grounded theoretical framework.
* Practical implementation requiring only black-box access to LLMs
* Comprehensive ablation studies that validate the contribution of each component

**Weaknesses:**

* The paper could benefit from more figures, particularly a good figure 1 to help explain the method. Possibly some ROC figures in the main paper since AUROC is the main metric used.

**Questions:**

* Could this method be used for open-ended generation tasks or is it only suitable for multiple choice questions?
* The same model seems to be used to both generating the explanations and evaluating their entailment probability. Doesn’t this introduce circularity? A model that has learned an incorrect fact could use that fact in an invalid explanation and then give a high entailment probability for that explanation.

---

> ### Author Response · Authors · 2024-11-28
> **Response to Reviewer 4HSJ**
>
> We thank the reviewer for their acknowledgement of strong empirical results and helpful feedback. We hope to address some concerns below, and have also provided **additional experimental results/figures** in the overall comment.
>
> **Using the same model for generation+evaluation:** Yes the same model is used for generation and evaluation, and you are certainly correct that if the model was trained on incorrect facts it may mistakenly place high entailment probability on an explanation. However, we would contend that the model’s uncertainty in this case should still be low, since our method is intended to measure the uncertainty of a specific model with respect to the logical relations it has learned during training.
>
> **Open Ended Generation:** This method has a natural extension to open-ended generation, please see the overall comment for our discussion.

---

### Official Review · Reviewer_EfjT · 2024-11-04

**Soundness:** 3
**Presentation:** 3
**Contribution:** 2
**Rating:** 6
**Confidence:** 4

**Summary:**

This paper addresses the task of confidence calibration in multi-choice QA setting, i.e., exploring how to quantify the likelihood that the answers provided by LLMs are correct.

The authors propose an ensemble-based method involving the following steps:
1. Explanation Sampling: Multiple explanations are sampled from the LLM using prompts (set with a temperature of 0.7) to generate diverse logical justifications for these different answers.
2. Explanation Probability Calculation: The probability associated with each explanation is calculated using the entailment prompt, i.e., prompting LLM to self-evaluate whether the explanation is correct given the question.
3. Marginal Probability Calculation: The overall answer confidence is calculated by computing the marginal probability of the answer token logits conditioned on each possible explanation.

The main contribution of this work is introducing a new ensemble-based mechanism that leverages explanations generated by the LLM itself. Experimental results show that this method achieves slight improvements over baseline approaches, particularly in complex reasoning tasks. However, its scope is primarily limited to multi-choice problems and includes only a few benchmark datasets​.

**Strengths:**

Originality: The paper proposes using explanations as latent variables to observe the model's uncertainty for each question and introduces a new ensemble method. This method evaluates the correctness of each generated explanation and then estimates answer confidence based on them. Compared to other methods such as consistency, it computes the final posterior probability of each answer based on all possible explanations. While the idea of using explanations may not be very new, the specific implementation and application of explanations in LLM's calibration setting has not been explored before.

Clarity: The paper is generally well-written and easy to follow.

Significance/Quality: Overall, the paper introduces a new ensemble mechanism and demonstrates, through experiments, that it shows marginal improvements over baseline methods. However, its impact is limited by the focus on the multi-choice setting and a small set of benchmarks, making the broader applicability and superiority of the method less evident. Future work could extend this approach to more diverse problem settings to further establish its value.

**Weaknesses:**

Clarity: One area for improvement is the articulation of the motivation behind using explanations for the ensemble method. Providing more depth on why explanations are suited for this purpose would enhance the reader’s understanding, especially why using explanation in this manner can mitigate overconfidence.

Experiment setting: I view this method as an ensemble-based calibration method, so I feel like it should be compared with more ensemble-based methods besides CoT-consistency.

Expense: The existing method's compuational complexity is O(n^2), while the CoT-consistency only requires O(n). I feel like the time complexity cost should be considered.

Overall significance: In general, this paper's impact is limited by the focus on the multi-choice setting and a small set of benchmarks, making the broader applicability and superiority of the method less evident. Future work could extend this approach to more diverse problem settings.

**Questions:**

1. Why did the authors choose to compute entailment probability and marginal probability for obtaining the final results, rather than performing semantic clustering on the explanations? The introduction in line 217 is too brief and somewhat difficult to understand. I feel like it would also be beneficial to try the semantic clustering approach during the experiment section or include a case study.
2. Why does utilizing explanations in this manner help mitigate overconfidence compared to directly using answers? The motivation for using explanation seems unclear to me, and I am a little bit struggled to understand why explanation can bring much more benefit than using multiple output answers directly.
3. In Figure 1a, why is there a need to condition on correct and incorrect answers instead of simply generating explanations based on the question?
4. The hyperlink in line 319 is invalid.
5. Why is the term "stable explanation" used?

---

> ### Author Response · Authors · 2024-11-28
> **Response to Reviewer EfjT**
>
> We thank the reviewer for the constructive comments and feedback. Please see the overall comment for discussion around the **stability terminology**, and **updated+additional experiments**. We address specific concerns as follows:
>
> **Computational Complexity + Semantic Clustering:** In the multiple choice setting, we make two additional calls (single generated token each) to the model for each explanation generated, one to evaluate entailment probability, and a second to determine the answer logit vector. As a result our method is still O(n) where n is the number of explanations sampled. This computational efficiency is also a reason why we did not initially use pairwise similarity between explanations. However, we agree that this would be an interesting line of future work.
>
> **Additional Experiments:** Please see the overall comment for additional experimental baselines, as well as an updated figure 1a which now examines explanations generated before the answer.
>
> **Ensemble-based calibration methods:** To our knowledge CoT-consistency is the primary ‘ensemble based’ method for use with LLMs without requiring access to internal weights or activations.
>
> **Additional Motivation for Using Explanations:** As for one concrete reason why explanation-based methods are better suited for these uncertainty tasks: consider that classical methods are restricted in evaluating uncertainty for a fixed set of hypothesis/answer classes. In contrast, our method can evaluate answers not seen during training. While using the LLM to predict something like P(True) [1] for this new answer will clearly not produce a calibrated likelihood that the answer is correct, we contend that the likelihood of answer+explanation pairs is still meaningful. This is because we make the assumption that while the train and test distributions may be different, both worlds follow the same underlying logic.
>
> For example, examining the default softmax of answer token probabilities for the following question on GPT-3.5 gives us:
>
> >Question: Carnows are gromulites. A busdriver is a carnow but not gorocks. Amy is a busdriver. What is Amy?
> >
> >A) gorocks → 0.034
> >
> >B) sherry → 0.039
> >
> >C) witherton → 0.013
> >
> >D) gromulite → 0.914
> >
>
> This model is obviously not calibrated for these new terms, as the clearly wrong answer (A) is still being assigned 3% probability of being correct. After conditioning on the correct logical explanation, this answer distribution becomes a singleton around the correct term (D).
>
> **Overall Significance:** Please see the overall comment for discussion around extending to our confidence metric to open-ended generation.
>
> ---
> [1] Kadavath, Saurav, et al. "Language models (mostly) know what they know." arXiv preprint arXiv:2207.05221 (2022).

---

### Official Review · Reviewer_dNm6 · 2024-11-04

**Soundness:** 2
**Presentation:** 2
**Contribution:** 2
**Rating:** 5
**Confidence:** 4

**Summary:**

This paper proposes a new method for measuring the confidence of LLM answers. The key idea of the method is to estimate the distribution of LLM answers by marginalizing over the distribution of explanations given for each answer. And instead of using the empirical explanation distribution, the authors reweight sampled explanations by the probability that they are entailed by the question. The authors evaluate their method on five question answering datasets and two LLMs (both GPT models) and find that it outperforms baselines in terms of selective uncertainty, but not calibration.

**Strengths:**

1. **Interesting and novel approach to LLM uncertainty estimation.** I think the idea of accounting for the accuracy/plausibility of the explanation when computing uncertainty estimates is interesting and intuitive. However, I think in the current version of the paper, this key idea gets lost in a lot of jargon/details (e.g., the Bayesian framing). I think the paper would be improved if this idea was centered more.
2. **Strong results on the task of selective uncertainty.** The proposed method clearly outperforms baselines on this important task.
3. **Figure 1 is a nice, intuitive illustration of the benefits of the method.** I thought this was a great way to make the point that accounting explanation plausibility provides useful information about the probability that the resulting answer is correct.

**Weaknesses:**

1. **Bayesian framing of your method is confusing/misleading.** You frame the method as the computation of a posterior distribution, as is done in Bayesian statistics. While your proposed method certainly resembles this, I don’t think this description is fully accurate. In Bayesian analysis, the computation of a posterior distribution involves updating your prior beliefs about a distribution (i.e., the prior) based on new evidence (i.e., the likelihood). To do this, you apply Baye’s rule, multiplying the prior by the likelihood and dividing by the evidence. However, your analysis does not involve the inclusion of a prior. Because your approach is not exactly Bayesian, I found this framing (as well as your use of the term “posterior”) confusing. I think it is a bit misleading and makes it more difficult to understand your method. While I still think it could be useful to describe the connection between your method and Bayesian analysis (e.g., you can say that your approach is similar to computing a posterior predictive distribution), I don’t think this should be used as the main way of describing your method. Instead, your computation of the answer distribution by marginalizing over the explanation distribution can be seen as an application of the law of total probability.
2. **Point about generalization to out-of-distribution test data is not fully supported.** You claim that one of the main weaknesses of existing uncertainty quantification methods is that they assume that the training and test data come from the same distribution. And you suggest that your method can overcome this weakness through its consideration of the explanation distribution. This is an interesting point that I believe is worth investigating, but it seems mainly speculative – the paper doesn’t provide any clear evidence to support this. In particular, it’s not fully clear to me that just because including an explanation as an intermediate step results in a new answer distribution, this new answer distribution will be better calibrated for the test data. And you don’t provide any sort of theoretical analysis to back this point. Further, your experiments are all on common benchmarks that could have been included in the GPT training data. Finally, it appears to me that the existing methods for uncertainty quantification based on CoT-consistency share the same benefit as your method with respect to this issue – they also consider explanations and therefore, “adapt” to the test data. However, you make it sound all existing methods make the ID assumption and therefore are harmed by differences between training and test data. Can you clarify this?
3. **Calibration performance is limited, and it’s not fully clear why we should discount this task compared to selective uncertainty.** The proposed method slightly underperforms some of the other methods on the ECE metric. I think this okay, given the improved performance in the selective uncertainty metrics. However, I am not fully convinced by the authors’ argument that this metric should be discounted. In the discussion section, the authors say that the ECE metric relies on the “hypothesis that the training, test, and calibration data are all drawn from the same distribution.” It is not clear to me why this is the case. I get that calibration methods rely on this assumption, but I don’t get why the *evaluation* of ECE relies on this assumption. Further, to account for the limitations around the effect of binning size, it seems like the authors could present results for multiple bin sizes or also present results for an alternative calibration metrics such as proposed in [1].
4. **Use of the term ‘stable’ is not clear.** You say that prompting a model to produce explanations, such as with CoT, “serves to ‘stabilize’ the sampled answers around an isolated minimum.” It is not totally clear to me what you mean by this, and at least in my reading of Wei et al., 2022, I don’t see this claim made. What exactly do you mean by this? Do you mean that prompting via CoT results in a lower entropy answer distribution? And do you have evidence to support this? I am mainly picking on this point because you term your method “stable” explanations, so I think it’s important that a reader can understand what you mean by “stable.”
5. **Writing lacks clarity.** This is a little bit redundant with the points made above, but in many places, I found that this paper was not fully clear, e.g.,:
* To describe your method in the intro, you say that it can be thought of as “computing the posterior predictive distribution by transductive marginalization over test-time classifiers.” This uses a lot of jargon, most of which you have not yet defined at this point (and some of which you never define, such as “transductive”). Also, as I said earlier, while what you do resembles computing a posterior predictive distribution, it isn’t the same in a strict sense. I think there are much simpler ways of describing your method, which would make the paper clearer.
* As described above, it is not clear what makes an explanation “stable.”
* In the methods section, you describe a step of your method that involves grouping explanations into equivalence classes by clustering them according to their resulting next token distributions. This step is unclear to me. Exactly how do you form these clusters? And then how do you use the equivalence classes in your computation of uncertainty?
* In your description of your method, I think it would be helpful to distinguish between the probability that an LLM outputs an answer (or explanation) from the probability that an answer (or explanation) is correct. Your current writing seems to conflate these two, which I think makes the text a bit harder to parse.
6. **Analysis is limited to GPT models.** It would strengthen the paper to see how the method works on other LLMs that are not GPT/OpenAI models (e.g., Llama, Claude, etc.).

[1] https://openaccess.thecvf.com/content_CVPRW_2019/papers/Uncertainty%20and%20Robustness%20in%20Deep%20Visual%20Learning/Nixon_Measuring_Calibration_in_Deep_Learning_CVPRW_2019_paper.pdf

**Questions:**

All my questions are in the weaknesses section.

---

> ### Author Response · Authors · 2024-11-28
> **Response to Reviewer dNm6**
>
> We thank the reviewer for their careful reading of the paper and detailed response. In the main comment we discuss additional **adaptive calibration metric results** as well as **stability and transductive terminology**. We hope to address additional concerns below:
>
> **Bayesian Clarification**: We thank the reviewer for this point about the Bayesian connection, and would like to add further clarification: if one considers the question itself as the evidence, we are computing a posterior likelihood of the explanation given the question using a Monte Carlo approach. Our explanation prior is implicit to the language model that we sample from: for example longer explanations will tend to be less likely and as a result sampled less often. With this in mind we have rewritten equation (5) to make the bayesian connection more explicit:
> \begin{equation}
>     p(e|q)=\frac{p(q|e)p(e)}{p(q)}\approx \frac{\phi_{ent.}(q,e)}{\sum_{e'\in E}\phi_{ent.}(q, e')} =: \hat{p}(e|q),
> \end{equation}
> We also state that we are calculating the ‘posterior predictive distribution’ on line 173 but will update references to the ‘Bayesian posterior’ to be more clear on lines 218 and 461.
>
> **Generalization to out-of-distribution test data:**
> * As noted by reviewer b15e, though somewhat standard the datasets selected are still “challenging” to LLMs, as indicated by lower overall accuracy. Additionally the GPT-4 technical report (Figure 8) suggests poor default calibration with datasets such as MMLU after applying post-training [1].
> * As for one reason why explanation based methods can better generalize: consider that classical methods are restricted in evaluating uncertainty for a fixed set of hypothesis/answer classes. In contrast, our method can evaluate answers not seen during training. While using the LLM to predict something like P(True) [2] for this new answer will clearly not produce a calibrated likelihood that the answer is correct, we contend that the likelihood of answer+explanation pairs is still meaningful. This is because we make the assumption that while the train and test distributions may be different, both worlds follow the same underlying logic.
> * You are correct that CoT-consistency also “adapts” to test data, however it is not guaranteed that this adaptation remains faithful to the data in the question [3].  Our method better ensures we are making a correct adaptation to the question and not just producing an explanation that drives the decision entropy to zero.
>
>
> **Additional Edits for Clarification:**
> * We have updated the mention of explanation clustering (line 216) to be more clear: instead of directly clustering explanations based on their pairwise similarity, we are effectively clustering based on their resulting answer distribution. If two explanations support the same answer, then they can be viewed as belonging to the same cluster. Our confidence for a specific answer is calculated by marginalizing over this cluster, i.e. all explanations that have non-zero probability mass assigned to that answer.
> * We have added additional clarification to distinguish between token level probability $p_\phi(a|q)$ and our estimate that answer is correct $\hat{p}(a|q)$.
>
> **Exploring Additional Models** Please see overall comment for discussion.
>
> ---
>
> [1] Achiam, Josh, et al. "Gpt-4 technical report." arXiv preprint arXiv:2303.08774 (2023).
>
> [2] Kadavath, Saurav, et al. "Language models (mostly) know what they know." arXiv preprint arXiv:2207.05221 (2022).
>
> [3] Turpin, Miles, et al. "Language models don't always say what they think: unfaithful explanations in chain-of-thought prompting." Advances in Neural Information Processing Systems 36 (2024).

---

> > ### Comment · Reviewer_dNm6 · 2024-12-03
> >
> > I thank the authors for the time and effort spent responding to my review. I still have a number of outstanding concerns, so I have kept my score the same. I discuss each point below.
> >
> > **Bayesian clarification**. I still do not think the Bayesian connection is fully clear. You say that “our explanation prior is implicit to the language model that we sample from”. However, you sample explanations conditioned on the question, so you sample from $p(e|q)$ rather than the prior $p(e)$. Further, it is not clear why $p(q|e)p(e) \approx \phi_{ent.}(q, e)$.
> >
> > **Generalization to out-of-distribution data.** I thank the authors for providing additional intuition as to why your method might lead to better generalization than approaches that do not consider explanations. I think the example provided in the general comment is helpful. However, I still have concerns about the claims made in the paper regarding this point (which are very similar to the concerns raised by reviewer b15e). In particular, you claim that OOD generalization is a key benefit of your approach compared to existing methods. While you provide intuition as to why this *could* be the case, you do not provide any definitive evidence to support this point. In order to make such strong statements, either theoretical analysis or empirical results (i.e., experiments on OOD data) are necessary.
> >
> > In addition, you claim that your method is more likely to generalize OOD compared to CoT-consistency because for CoT-consistency “it is not guaranteed that this adaptation remains faithful to the data in the question [3].” I found this point confusing. It is not clear what you mean by “faithful to the data in the question.” Faithfulness as studied in [3] has to do with whether the explanation is consistent with the LLM’s underlying reasoning – not the “data in the question.” Faithfulness is different from logical entailment, which is what your method checks for. So to me, it is not clear that [3] is evidence that CoT-consistency will have worse OOD performance compared to your method.
> >
> > **Calibration performance.** I thank the authors for their work on including adaptive calibration error as an evaluation metric. However, I think the core concerns I expressed in my review regarding this point are still unaddressed. In particular, I am still not convinced that it makes sense to discount calibration so heavily in the paper (e.g., putting it in red). In particular, the authors have not yet clarified their claim that ECE relies on the “hypothesis that the training, test, and calibration data are all drawn from the same distribution.” It is still not clear to me why this is the case. Also, it is not clear that this criticism applies to the setting studied by the authors -- which is not OOD.
> >
> > **Additional models.** I am not fully convinced of the lack of need for experiments on additional models. While the authors mention several papers that show that certain LLM capabilities are similar across different models, none of these papers study uncertainty quantification. Therefore, it is not clear that the performance of uncertainty quantification methods is consistent across different models.
> >
> > **Clarity.** My concerns about the clarity of the paper still remain.
> > * To me, it doesn’t appear that any changes were made to clarify the explanation clustering step in the paper (line 216 appears identical in this version and the previous one). Can the authors please point me to exactly what changes were made? This step is still not clear to me – is the clustering an explicit step in the method or are you simply claiming that your method can be viewed as implicitly clustering explanations?
> > * Where have you added additional clarification to distinguish between token level probabilities and the estimate that the answer is correct? I do not see changes to this effect in the paper. Can you please point me to them?

---

### Author Response · Authors · 2024-11-28
**Overall Author Response and Key Updates to Manuscript (Part 1/2)**

**Dear Reviewers,**

Thank you for the careful reading of our paper. We have submitted an improved manuscript incorporating your constructive comments and feedback. Below, we summarize the key updates made:

**Improving introduction of Stable, Transductive Terminology:**

The ‘stability’ in our stable explanations method primarily comes from the fact that the answer distribution tends to collapse toward a single response after conditioning on an explanation; we have provided additional experiments (Appendix B4, Fig 9. of the revision) demonstrating this effect, with observed entropy decrease for $>75$% of examples across datasets. Our intent with this terminology is to highlight the implicit dynamics of the language model state as we see additional tokens in the explanation. Questions that the model is certain about should have explanation 'trajectories' toward these local answer minima while remaining logically consistent. We have added additional clarification to the introduction as well as a reference to the new figure (referenced in the introduction paragraph at line 51). We have also reworded the mentions in introduction of transductive inference, providing a clearer definition/reference and emphasizing the balance between learning model weights during training and reasoning chains during inference.

**Updated Section 4.2 Experiment:**

We thank reviewers  dNm6, EfjT, and b15e for the point about the order of conditioning in section 4.2. We have updated our experiment (Fig.1) so that the explanation is now generated before the answer (q→e→a matching the way explanations are generated in our stable confidence metric). We can observe slightly lower token likelihoods on average for explanations associated with incorrect answers, but still the distributions are very close. We also still see for the entailment probability a distinct tail for incorrect explanations that help us to rule out the least logically consistent explanations.

**Additional Baselines and Larger Sample Size:**

As suggested by reviewer b15e, we have increased sample size, doubling the number of samples for each experiment (250→500 for gpt3.5 and 100→200) and have updated the manuscript with the revised results (Figure 2). Average performance improvements have remained roughly the same, with slightly more consistency across specific benchmarks. We would like to highlight that our method is now beating baselines for *9/10* selective uncertainty metrics+benchmarks on GPT-4 (with the exception of AURC on MMLU Law).
Additionally we have added P(True) from Kadavath et al. 2022 as a new baseline, and have experiments in the Appendix demonstrating that semantic entropy [1] is almost exactly correlated with the answer token probability (Token Prob.) in the case of multiple choice questions.

**Moving to Adaptive Calibration Error:**

As suggested by reviewer dNm6, we have recalculated calibration error using a slightly different ‘adaptive’ formulation that groups samples into equally sized bins (as opposed to equal intervals). This adaptive calibration error [2] returns slightly better results for our method when evaluated on GPT-4, within 0.01 of the best baseline on average and beating the best baseline on 2/5 datasets. We also note that one can easily ‘calibrate’ any un-normalized confidence metric for a specific dataset through a simple monotonic transformation provided that incorrect and correct samples can be separated (i.e. high AUROC/AURC).

---

> ### Author Response · Authors · 2024-11-28
> **Overall Author Response and Key Updates to Manuscript (Part 2/2)**
>
> We provide additional discussion below for some common reviewer concerns and questions.
>
> ---
>
> **OOD Generalization:** One reason explanation-based methods can better generalize to unseen data is that classical methods are restricted in evaluating uncertainty for a fixed set of hypothesis/answer classes. Although one may naively have the LLM predict something like the probability that a new answer class is true (i.e. P(True) from [1]), this is not guaranteed to produce a calibrated likelihood that the answer is correct.
> In contrast, we contend that the likelihood of explanation+answer pairs is still meaningful. This is because we make the assumption that while the train and test distributions may be different, *both worlds follow the same underlying logic*.
>
> For example, examining the default softmax of answer token probabilities for the following question on GPT-3.5 yields:
>
> >Question: Carnows are gromulites. A busdriver is a carnow but not gorocks. Amy is a busdriver. What is Amy?
> >
> >A) gorocks → 0.034
> >
> >B) sherry → 0.039
> >
> >C) witherton → 0.013
> >
> >D) gromulite → 0.914
> >
>
> This model is obviously not calibrated for these invented terms, as the clearly wrong answer (A) is still being assigned 3% probability of being correct. After conditioning on a CoT explanation, this answer distribution becomes a singleton around the correct term (D). Our method attempts to select for these logical explanations and marginalize over their resulting answer distributions.
>
> **Open-ended Extension:** One natural extension of our method to open-ended generation would be to generate explanation+answer pairs, and then treat these answers as our new choices. We can then evaluate the likelihood of these choices given a specific explanation to determine the conditional answer distribution. This type of “cross-perplexity” evaluation has connections to semantic similarity metric proposed in [3] and from this perspective could be seen as marginalizing over the meaning of all logically entailed explanations associated with a question.
>
> **Additional Models:** We primarily focus on testing our method on models at different *scales* rather than across model families for two reasons. First, existing literature [4] has already shown the effect of CoT explanations to be similar across a variety of architectures. Additional work [5,6] has also shown that logical reasoning capability such as binary entailment is primarily a function of model size.  In particular, Parmar et al. 2024 [5] observe similar performance in logical reasoning for llama2-7b and mistral-7b models and observe consistent increases in performance as they scale to larger models such as Yi-34B and then GPT4.
>
> ---
> [1] Kuhn, Lorenz, Yarin Gal, and Sebastian Farquhar. "Semantic uncertainty: Linguistic invariances for uncertainty estimation in natural language generation." arXiv preprint arXiv:2302.09664 (2023).
>
> [2] Nixon, Jeremy, et al. "Measuring Calibration in Deep Learning." CVPR workshops. Vol. 2. No. 7. 2019.
>
> [3] Liu, Tian Yu, et al. "Meaning representations from trajectories in autoregressive models." arXiv preprint arXiv:2310.18348 (2023).
>
> [4] Wang, Xuezhi, et al. "Self-consistency improves chain of thought reasoning in language models." arXiv preprint arXiv:2203.11171 (2022).
>
> [5] Sanyal, Soumya, et al. "Minds versus Machines: Rethinking Entailment Verification with Language Models." arXiv preprint arXiv:2402.03686 (2024).
>
> [6] Parmar, Mihir, et al. "LogicBench: Towards systematic evaluation of logical reasoning ability of large language models." Proceedings of the 62nd Annual Meeting of the Association for Computational Linguistics (Volume 1: Long Papers). 2024.

---

### Meta-Review · Area_Chair_hQpB · 2024-12-21

**Metareview:**

The paper measures LLM confidence through stable explanations, leveraging the distribution of generated explanations and their entailment probabilities to estimate uncertainty. The reviewers acknowledged the novelty of using explanation-based reweighting but raised significant concerns regarding clarity, theoretical grounding, and experimental results. While additional experiments during the rebuttal improved clarity and introduced new baselines, the reviewers did not reach a consensus. I recommend rejection.

**Additional Comments On Reviewer Discussion:**

During the rebuttal, the authors addressed concerns about clarity and experimental robustness by providing additional baselines and refining their explanation of stable confidence measures. While these changes improved clarity and partially addressed reviewer concerns about scope and methodology, some key issues remained unresolved.

---

### Decision · Program_Chairs · 2025-01-22

Reject